# Variational Inequality Methods for Multi-Agent Reinforcement Learning: Performance and Stability Gains

## Abstract

Multi-agent reinforcement learning (MARL) poses distinct challenges as agents learn strategies through experiences. Gradient-based methods often fail to converge in MARL, and performances are highly sensitive to initial random seeds, contributing to what has been termed the *MARL reproducibility crisis*. Concurrently, significant advances have been made in solving Variational Inequalities (VIs)—which include equilibrium-finding problems—particularly in addressing the non-converging rotational dynamics that impede convergence of traditional gradient-based optimization methods. This paper explores the potential of leveraging VI-based techniques to improve MARL training. Specifically, we study the integration of VI methods—namely, Nested-Lookahead VI (nLA-VI) and Extragradient (EG)—into the *multi-agent deep deterministic policy gradient* (MADDPG) algorithm. We present a VI reformulation of the actor-critic algorithm for both single- and multi-agent settings. We introduce three algorithms that use nLA-VI, EG, and a combination of both, named *LA-MADDPG*, *EG-MADDPG*, and *LA-EG-MADDPG*, respectively. Our empirical results show that these VI-based approaches yield significant performance improvements in benchmark environments, such as the zero-sum games: *rock-paper-scissors and matching pennies*, where equilibrium strategies can be quantitatively assessed, and the *Multi-Agent Particle Environment: Predator-prey* benchmark, where VI-based methods also yield balanced participation of agents from the same team, further highlighting the substantial impact of advanced optimization techniques on MARL performance.

## 1 Introduction

Multi-agent reinforcement learning (MARL) is a powerful machine learning approach for solving complex, multi-player problems in diverse domains. It has been applied to tasks such as coordinating multi-robot and multi-drone systems for instance for search and warehouse automation, optimizing traffic flow and vehicle platooning in autonomous driving, managing energy distribution in smart grids, simulating financial markets and automated trading, improving patient management and drug discovery in healthcare, enhancing network performance in telecommunications, training intelligent agents in games (e.g., Omidshafiei et al., 2017; Vinyals et al., 2017; Spica et al., 2018; Zhou et al., 2021; Bertsekas, 2021, among others). In MARL, a system of $N$ agents seeks to jointly optimize a shared objective, with each agent operating based on its own policy, derived from its observations of the environment. Depending on their reward objectives, agents may exhibit cooperative, competitive, or mixed behaviors. These interactions introduce complex learning dynamics, making MARL significantly more challenging and distinct from single and actor-only reinforcement learning (RL).

Despite the applicability of MARL to a wide range of problems, their deployment and research development face significant challenges. Key issues include: *(i)* The iterative training process in data-driven MARL is notoriously difficult, often failing to start to converge. This lack of convergence remains a fundamental obstacle. *(ii)* Performance is highly sensitive to small changes in hyperparameters or initial random seed variations, leading to unpredictable outcomes. This hinders reproducibility. These challenges are also evident in single-agent reinforcement learning with actor-critic structure (Konda & Tsitsiklis, 1999)–an instance of a two-player game. Such methods require meticulous hyperparameter tuning, and their outcomes can vary significantly based on the

random seeds used for sampling and model initialization (Wang et al., 2022; Eimer et al., 2023). This variability undermines reproducibility, complicating both research progress and real-world deployment (Henderson et al., 2019; Lynnerup et al., 2019). In MARL, these challenges are even more pronounced, contributing to what is often referred to as the *reproducibility crisis* (Bettini et al., 2024). Small changes to hyperparameter values can drastically alter results, as demonstrated by Gorsane et al. (2022). Their study reveals significant performance variability across different seeds in popular MARL benchmarks such as the *StarCraft* multi-agent challenge (Samvelyan et al., 2019). Additionally, gradient-based optimization methods in MARL face unique challenges, such as difficulties in exploring the joint policy space of multiple agents (Li et al., 2023; Christianos et al., 2021), often leading to suboptimal solutions. Some MARL structures also exhibit inherent cycling effects (Zheng et al., 2021), further exacerbating the problem of convergence. In this work, we focus primarily on addressing challenge (i) by improving training stability. In doing so, we also aim to mitigate the high variability caused by random seed and hyperparameter sensitivity, thereby partially addressing (ii).

A concurrent line of works focuses on the *Variational Inequality* (VI) problem, a general class of problems that encompasses both equilibria- and optima-finding problems; see Section 3 for definition. VIs generalize standard minimization problems. In this case, the operator $F$ is a gradient field $F \equiv \nabla f$. However, by allowing $F$ to be a more general vector field, VIs also model problems such as finding equilibria in zero-sum and general-sum games (Cottle & Dantzig, 1968; Rockafellar, 1970). It has been observed that standard optimization methods that perform well in minimization tasks of the form $\min_{z} f(z)$—where $f \colon \mathbb{R}^d \to \mathbb{R}$ is a real-valued loss function—often fail to solve some simple instances of VIs. This failure is due to the *rotational* component inherent in the gradient dynamics of these settings, which causes the latest iterate to cycle around the solution, leading to non-convergence (Mescheder et al., 2018; Balduzzi et al., 2018). More precisely, the Jacobian of the associated vector field (see def. in Section 3) can be decomposed into a symmetric and antisymmetric component (Balduzzi et al., 2018), where each behaves as a *potential* (Monderer & Shapley, 1996) and a *Hamiltonian* (purely rotational) game, resp. For instance, the gradient descent method for the simple $\min_{z_1 \in \mathbb{R}^{d_1}} \max_{z_2 \in \mathbb{R}^{d_2}} \; z_1 \cdot z_2$ game, which simultaneously updates $z_1, z_1$ rotates around the solution for infinitesimally small learning rates, and diverges away from it for practical choices of its value. As a result, all variation methods based on gradient descent, such as *Adam* (Kingma & Ba, 2015) have no hope of converging for some broad problem classes of VIs. This problematic behavior is particularly pronounced when the separate sets of parameters are neural networks, as in generative adversarial networks (GANs, Goodfellow et al., 2014). As a result, when GANs were first introduced, substantial computational resources were required to fine-tune hyperparameters (Radford et al., 2016). In addition, even for highly tuned hyperparameters, training with minimization methods often (eventually) diverges away (Chavdarova et al., 2021), in sharp contrast to standard minimization. The original GAN formulation (Goodfellow et al., 2014) and practical GAN implementations that are not necessarily zero-sum, are all instances of VIs. The above training difficulties inspired numerous recent research efforts to develop numerical methods to approximately solve variational inequalities (VIs) and to study how VI optimization differs from minimization. Various algorithms have been proposed and studied; reviewed in Sections 2 and 3 and Appendix A.1.1.

This paper hypothesizes that training instabilities in MARL and actor-critic RL primarily arise from the rotational dynamics inherent in competitive learning objectives. This issue is compounded by the prevalent reliance in practice on gradient descent-based methods, such as Adam (Kingma & Ba, 2015), which are known to induce similar issues in the Variational Inequality (VI) literature and have been observed to cause analogous challenges in other VI applications (Goodfellow et al., 2014; Gidel et al., 2019a; Chavdarova et al., 2021). Since VI optimization methods are specifically designed to address rotational vector fields, this paper raises the following question:

*Can MARL algorithms benefit from the application of Variational Inequality optimization methods?*

To address this question, we focus on the *multi-agent deep deterministic policy gradient* (MADDPG) method (Lowe et al., 2017) and integrate it with the (combination of) *nested-Lookahead-VI* (nLA-VI) (Chavdarova et al., 2021) and *Extragradient* (EG) (Korpelevich, 1976) methods for solving variational inequalities (VIs). Our main contributions are as follows:

- Primerely, we present a VI perspective for multi-agent reinforcement learning (MARL) problems.

- We propose the *LA-MADDPG, EG-MADDPG*, and *LA-EG-MADDPG* algorithms, which extend MADDPG by combining it with nLA-VI, and with EG and a mix of both (respectively) in the actor-critic parameter optimization for all agents.

- We evaluate the proposed methods against standard optimization approaches in several two-player games and benchmarks from the Multi-Agent Particle Environment (MPE, Lowe et al., 2017).

- We also discuss additional insights into the use of rewards as a performance metric in MARL.

## 2 RELATED WORKS

Our work draws mainly from two lines of work that we review next.

**Multi-Agent Reinforcement Learning (MARL).** Various MARL algorithms have been developed (Lowe et al., 2017; Iqbal & Sha, 2018; Ackermann et al., 2019; Yu et al., 2021), with some extending existing single-agent reinforcement learning (RL) methods (Rashid et al., 2018; Son et al., 2019; Yu et al., 2022; Kuba et al., 2022). Lowe et al. (2017) extend the actor-critic algorithm to the MARL setting using the *centralized training decentralized execution* framework. In the proposed algorithm, named *multi-agent deep deterministic policy gradient (MADDPG)*, each agent in the game consists of two components: an *actor* and a *critic*. The actor is a policy network that has access only to the local observations of the corresponding agent and is trained to output appropriate actions. The critic is a value network that receives additional information about the policies of other agents and learns to output the Q-value; see Section 3. After a phase of experience collection, a batch is sampled from a replay buffer and used for training the agents. To our knowledge, all deep MARL implementations rely on either stochastic gradient descent or *Adam* optimizer (Kingma & Ba, 2015) to train all networks. Game theory and MARL share many foundational concepts, and several studies explore the relationships between the two fields (Yang & Wang, 2021; Fan, 2024), with some using game-theoretic approaches to model MARL problems (Zheng et al., 2021). This work proposes incorporating game-theoretic techniques into the optimization process of existing MARL methods to determine if these techniques can enhance MARL optimization.

**Variational Inequalities (VIs).** VIs were first formulated to understand the equilibrium of a dynamical system (Stampacchia, 1964). Since then, they have been studied extensively in mathematics including operational research and network games (see Facchinei & Pang, 2003, and references therein). More recently, after the shown training difficulties of GANs (Goodfellow et al., 2014)—which are an instance of VIs—an extensive line of works in machine learning studies the convergence of iterative gradient-based methods to solve VIs numerically. Since the last and average iterates can be far apart when solving VIs (see e.g., Chavdarova et al., 2019), these works primarily aimed at obtaining last-iterate convergence for special cases of VIs that are important in applications, including bilinear or strongly monotone games (e.g., Tseng, 1995; Malitsky, 2015; Facchinei & Pang, 2003; Daskalakis et al., 2018; Liang & Stokes, 2019; Gidel et al., 2019b; Azizian et al., 2020; Thekumparampil et al., 2022), VIs with cocoercive operators (Diakonikolas, 2020), or monotone operators (Chavdarova et al., 2023; Gorbunov et al., 2022). Several works *(i)* exploit continuous-time analyses (Ryu et al., 2019; Bot et al., 2020; Rosca et al., 2021; Chavdarova et al., 2023; Bot et al., 2022), *(ii)* establish lower bounds for some VI classes (e.g., Golowich et al., 2020b;a), and *(iii)* study the constrained setting (Daskalakis & Panageas, 2019; Cai et al., 2022; Yang et al., 2023; Chavdarova et al., 2024), among other. Due to the computational complexities involved in training neural networks, iterative methods that rely solely on first-order derivative computation are the most commonly used approaches for solving variational inequalities (VIs). However, standard gradient descent and its momentum-based variants often fail to converge even on simple instances of VIs. As a result, several alternative methods have been developed to address this issue. Some of the most popular first-order methods for solving VIs include the *extragradient* method (Korpelevich, 1976), *optimistic gradient* method (Popov, 1980), *Halpern* method (Diakonikolas, 2020), and (nested) *Lookahead-VI* method (Chavdarova et al., 2021); these are discussed in detail in Section 3 and Appendix A.1.1. In this work, we primarily focus on the nested Lookahead-VI method, which has achieved state-of-the-art results on the CIFAR-10 (Krizhevsky, 2009) benchmark for generative adversarial networks (Goodfellow et al., 2014).

## 3 PRELIMINARIES

**Notation.** Bold small letters denote vectors, and curly capital letters denote sets. Let $\mathcal{Z}$ be a convex and compact set in the Euclidean space, with inner product $\langle \cdot, \cdot \rangle$.

**Setting: multi-agent deep deterministic policy gradient.** *Markov Games* (MGs) extend Markov Decision Processes to the multi-agent setting. In a Markov Game, $N$ agents interact within an environment characterized by a set of states $\mathcal{S}$. Agents receive observations $\boldsymbol{o}_i, i = 1, \ldots, N$ of the current environment state $\boldsymbol{s} \in \mathcal{S}$. Based on their policies $\pi_i$, each agent $i$ chooses an action $a_i \in \mathcal{A}_i$ from predefined finite action sets $\mathcal{A}_i, i = 1, \ldots, N$. These actions, collectively represented as $\mathbf{a}$, are then applied to the environment, which transitions to a new state $\hat{\boldsymbol{s}} \in \mathcal{S}$ according to a transition function $\mathcal{T} \colon \mathcal{S} \to \mathcal{S}$. Each agent receives a reward $r_i, i = 1 \ldots N$, and a new observation $\hat{\boldsymbol{o}}_i$. In the MARL setting herein, each agent has its own Q-value that is, how much reward it expects to get from a state when joint action $\mathbf{a}$ is performed.

*Multi-agent deep deterministic policy gradient (MADDPG, Lowe et al., 2017)*, extends Deep deterministic policy gradient (DDPG, Lillicrap et al., 2015) to multi-agent setting using the framework of *centralized training decentralized execution*. Each agent $i$ has (i) a *critic network*—$Q_i$—which acts as a centralized action-value function, (ii) a target critic network $Q_i'$ that is less frequently updated with the most recent $Q_i$ parameters for learning stability, (iii) an *actor network* $\mu_i$ which represents the policy to be updated, and (iv) a target actor network $\mu_i'$ from which it selects its actions and is periodically updated with the learned policy $\mu_i$. Both the Critic and Actor networks are modeled using feedforward networks, parameterized by $\boldsymbol{w}$ and $\boldsymbol{\theta}$ respectively.

**The VI framework.** Broadly speaking, VIs formalize equilibrium-seeking problems. The goal is to find an equilibrium $\boldsymbol{z}^\star$ from the domain of continuous strategies $\mathcal{Z}$, such that:

$$\langle \boldsymbol{z} - \boldsymbol{z}^\star, F(\boldsymbol{z}^\star) \rangle \geq 0, \quad \forall \boldsymbol{z} \in \mathcal{Z}, \tag{VI}$$

where the so-called *operator* $F \colon \mathcal{Z} \to \mathbb{R}^n$ is continuous, and $\mathcal{Z}$ is a subset of the Euclidean $d$-dimensional space $\mathbb{R}^d$. Thus, VIs are defined by the tuple $F, \mathcal{Z}$, denoted herein as VI$(F, \mathcal{Z})$. This problem is equivalent to standard minimization, when $F \equiv \nabla f$, where $f$ is a real-valued function $f \colon \mathbb{R}^d \to \mathbb{R}$. We refer the reader to (Facchinei & Pang, 2003) for an introduction and examples. To illustrate the relevance of VIs to multi-agent problems, consider the following example. Suppose we have $N$ agents, each with a strategy $\boldsymbol{z}_i \in \mathbb{R}^{d_i}$, and let us denote the joint strategy with $\boldsymbol{z} \equiv [\boldsymbol{z}_1^\intercal, \ldots, \boldsymbol{z}_N^\intercal]^\intercal \in \mathbb{R}^d, d = \sum_{i=1}^N d_i$. Each agent aims to optimize its objective $f_i \colon \mathbb{R}^d \to \mathbb{R}$. Then, finding an equilibrium in this game is equivalent to solving a VI where $F$ corresponds to:
$F(\boldsymbol{z}) \equiv [\nabla_{\boldsymbol{z}_1} f_1(\boldsymbol{z}), \ldots, \nabla_{\boldsymbol{z}_N} f_N(\boldsymbol{z})]^\intercal$.

**Methods for solving VIs.** The *gradient descent* method naturally extends for the VI problem as follows:

$$\boldsymbol{z}_{t+1} = \boldsymbol{z}_t - \eta F(\boldsymbol{z}_t), \tag{GD}$$

where $t$ denotes the iteration count, and $\eta \in (0, 1)$ the step size or learning rate. The *nested-Lookahead-VI* algorithm for VI problems (Chavdarova et al., 2021), originally proposed for minimization by Zhang et al. (2019), is a general wrapper of a "base" optimizer where, at every step $t$: (i) a copy of the current iterate $\tilde{\boldsymbol{z}}_t$ is made: $\tilde{\boldsymbol{z}}_t \leftarrow \boldsymbol{z}_t$, (ii) $\tilde{\boldsymbol{z}}_t$ is updated $k \geq 1$ times, yielding $\tilde{\boldsymbol{\omega}}_{t+k}$, and finally (iii) the actual update $\boldsymbol{z}_{t+1}$ is obtained as a *point that lies on a line between* the current $\boldsymbol{z}_t$ iterate and the predicted one $\tilde{\boldsymbol{z}}_{t+k}$:

$$\boldsymbol{z}_{t+1} \leftarrow \boldsymbol{z}_t + \alpha(\tilde{\boldsymbol{z}}_{t+k} - \boldsymbol{z}_t), \quad \alpha \in [0, 1]. \tag{(nested)LA-VI}$$

Notice that we can apply this idea recursively, and when the base optimizer is (nested)LA-VI (at some level), then we have *nested* LA-VI, as proposed in Algorithm 3 in (Chavdarova et al., 2021).

*Extragradient* (Korpelevich, 1976) uses a "prediction" step to obtain an extrapolated point $\boldsymbol{z}_{t+\frac{1}{2}}$ using GD: $\boldsymbol{z}_{t+\frac{1}{2}} = \boldsymbol{z}_t - \eta F(\boldsymbol{z}_t)$, and the gradients at the *extrapolated* point are then applied to the *current* iterate $\boldsymbol{z}_t$ as follows:

$$\boldsymbol{z}_{t+1} = \boldsymbol{z}_t - \eta F\Big(\boldsymbol{z}_t - \eta F(\boldsymbol{z}_t)\Big), \tag{EG}$$

where $\eta > 0$ is a learning rate (step size). Unlike gradient descent, EG converges on some simple game instances, such as in games linear in both players (Korpelevich, 1976).

---

**Algorithm 1** Procedure Pseudocode for nLA-VI, called from Algorithm 2.

---

1: **procedure** NESTEDLOOKAHEAD**:**
2:     **Input:** Number of agents $N$, current episode $e$, current actor weights and snapshots $\{(\boldsymbol{\theta}_i, \boldsymbol{\theta}_{i,s}, \boldsymbol{\theta}_{i,ss},)\}_{i=1}^N$, current critic weights and snapshots $\{(\boldsymbol{w}_i, \boldsymbol{w}_{i,s}, \boldsymbol{w}_{i,ss})\}_{i=1}^N$, lookahead hyperparameters $k_s$, $k_{ss}$ (where $k_{ss}$ can be $\varnothing$) and $\alpha_{\boldsymbol{\theta}}, \alpha_{\boldsymbol{w}}$
3:     **Result:** Updated actor and critic weights and snapshots for all agents
4:     **if** $e\%k_s == 0$ **then**
5:         **for all** agent $i \in 1, \ldots, N$ **do**
6:             $\boldsymbol{w}_i \leftarrow \boldsymbol{w}_{i,s} + \alpha_{\boldsymbol{w}}(\boldsymbol{w}_i - \boldsymbol{w}_{i,s})$                         *Apply lookahead (1st level)*
7:             $\boldsymbol{\theta}_i \leftarrow \boldsymbol{\theta}_{i,s} + \alpha_{\boldsymbol{\theta}}(\boldsymbol{\theta}_i - \boldsymbol{\theta}_{i,s})$
8:             $(\boldsymbol{\theta}_{i,s}, \boldsymbol{w}_{i,s}) \leftarrow (\boldsymbol{\theta}_i, \boldsymbol{w}_i)$                         *Update snapshots (1st level)*
9:         **end for**
10:     **end if**
11:     **if** $k_{ss}$ is *not* $\varnothing$ and $e\%k_{ss} == 0$ **then**
12:         **for all** agent $i \in 1, \ldots, N$ **do**
13:             $\boldsymbol{w}_i \leftarrow \boldsymbol{w}_{i,ss} + \alpha_{\boldsymbol{w}}(\boldsymbol{w}_i - \boldsymbol{w}_{i,ss})$                     *Apply lookahead (2nd level)*
14:             $\boldsymbol{\theta}_i \leftarrow \boldsymbol{\theta}_{i,ss} + \alpha_{\boldsymbol{\theta}}(\boldsymbol{\theta}_i - \boldsymbol{\theta}_{i,ss})$
15:             $(\boldsymbol{\theta}_{i,s}, \boldsymbol{\theta}_{i,ss}, \boldsymbol{w}_{i,s}, \boldsymbol{w}_{i,ss}) \leftarrow (\boldsymbol{\theta}_i, \boldsymbol{\theta}_i, \boldsymbol{w}_i, \boldsymbol{w}_i)$     *Update snapshots (1st & 2nd level)*
16:         **end for**
17:     **end if**
18: **end procedure**

---

## 4 A VI PERSPECTIVE & OPTIMIZATION METHODS FOR MARL

Herein, we describe our proposed approach, which utilizes VI methods in combination with a MARL algorithm. Specifically, we delve into MADDPG, give a VI perspective of it, and describe its combination with extragradient (Korpelevich, 1976) and nested Lookahead (Chavdarova et al., 2021).

### 4.1 A VI PERSPECTIVE OF MADDPG

Recall that for each $i = 1, \ldots, N$ agent, we have:

1. $Q$–Network, $\mathbf{Q}_i^{\boldsymbol{\mu}}(\mathbf{x}, a_1, \ldots, a_N; \boldsymbol{w}_i)$: central critic network for agent $i$;

2. Policy network, $\boldsymbol{\mu}_i(o_i; \boldsymbol{\theta}_i)$: policy network for agent $i$;

3. Target $Q$–network, $\mathbf{Q}_i^{\boldsymbol{\mu}'}(\mathbf{x}, a_1, \ldots, a_N; \boldsymbol{w}_i')$;

4. Target policy network, $\boldsymbol{\mu}_i'(o_i; \boldsymbol{\theta}_i')$.

These networks (maps) are parametrized by $\boldsymbol{w}_i, \boldsymbol{\theta}_i, \boldsymbol{w}_i', \boldsymbol{\theta}_i'$, respectively; with $\boldsymbol{w}_i, \boldsymbol{w}_i' \in \mathbb{R}^{d_i^Q}$ and $\boldsymbol{\theta}_i, \boldsymbol{\theta}_i' \in \mathbb{R}^{d_i^{\mu}}$. The latter two—$\boldsymbol{w}_i', \boldsymbol{\theta}_i'$ for agent $i$—are running averages computed as:

$$
\begin{aligned}
\boldsymbol{\theta}_i' &\leftarrow \tau\boldsymbol{\theta}_i + (1-\tau)\boldsymbol{\theta}_i' \\
\boldsymbol{w}_i' &\leftarrow \tau\boldsymbol{w}_i + (1-\tau)\boldsymbol{w}_i'
\end{aligned}
. \tag{Target-Nets}
$$

Given a batch of experiences $(\mathbf{x}^j, \mathbf{a}^j, \mathbf{r}^j, \hat{\mathbf{x}}^j)$—sampled from a replay buffer $(\mathcal{D})$—the goal is to find an equilibrium by solving the VI problem with the operator $F$ defined as:

$$
F_{\text{MADDPG}}\left(\begin{bmatrix} \vdots \\ \boldsymbol{w}_i \\ \boldsymbol{\theta}_i \\ \vdots \end{bmatrix}\right) \equiv \begin{bmatrix} \vdots \\ \nabla_{\boldsymbol{w}_i} \frac{1}{S} \sum_j \left( r_i^j + \gamma \mathbf{Q}_i^{\boldsymbol{\mu}'}\left(\hat{\mathbf{x}}^j, a_1', \ldots, a_N'; \boldsymbol{w}_i'\right)\big|_{a_k' = \boldsymbol{\mu}_k'(o_k^j)} - \mathbf{Q}_i^{\boldsymbol{\mu}}(\mathbf{x}^j, \mathbf{a}^j; \boldsymbol{w}_i) \right)^2 \\ \frac{1}{S} \sum_j \nabla_{\boldsymbol{\theta}_i} \boldsymbol{\mu}_i(o_i^j; \boldsymbol{\theta}_i) \nabla_{a_i} \mathbf{Q}_i^{\boldsymbol{\mu}}(\mathbf{x}^j, a_1^j, \ldots, a_i, \ldots, a_N^j; \boldsymbol{w}_i)\big|_{a_i = \boldsymbol{\mu}_i(o_i^j)} \\ \vdots \end{bmatrix},
\tag{$F_{\text{MADDPG}}$}
$$

and $\mathcal{Z} \equiv \mathbb{R}^d$, where $d = \sum_{i=1}^N (d_i^Q + d_i^{\mu})$. Even if $N = 1$, there is still a game between the actor and critic—the update of $\boldsymbol{w}_i$ depends on $\boldsymbol{\theta}_i$ and vice versa.

## 4.2 PROPOSED METHODS

To solve the VI problem with the operator as defined in ($F_{\mathrm{MADDPG}}$), we propose the *LA-MADDPG*, and *EG-MADDPG* methods, described in detail in this section.

---

**Algorithm 2** Pseudocode for LA–MADDPG: MADDPG with (Nested)-Lookahead-VI.

---

1: **Input:** Environment $\mathcal{E}$, number of agents $N$, number of episodes $T$, action spaces $\{\mathcal{A}_i\}_{i=1}^{N}$, random steps $T_{\mathrm{rand}}$, learning interval $T_{\mathrm{learn}}$, actor networks $\{\boldsymbol{\mu}_i\}_{i=1}^{N}$ with weights $\boldsymbol{\theta} \equiv \{\boldsymbol{\theta}_i\}_{i=1}^{N}$, critic networks $\{\mathbf{Q}_i\}_{i=1}^{N}$ with weights $\boldsymbol{w} \equiv \{\boldsymbol{w}_i\}_{i=1}^{N}$, target actor networks $\{\boldsymbol{\mu}_i'\}_{i=1}^{N}$ with weights $\boldsymbol{\theta}' \equiv \{\boldsymbol{\theta}_i'\}_{i=1}^{N}$, target critic networks $\{\mathbf{Q}_i'\}_{i=1}^{N}$ with weights $\boldsymbol{w}' \equiv \{\boldsymbol{w}_i'\}_{i=1}^{N}$, learning rates $\eta_{\boldsymbol{\theta}}, \eta_{\boldsymbol{w}}$, optimizer $\mathcal{B}$, discount factor $\gamma$, lookahead parameters $k_s, k_{ss}, \alpha_{\boldsymbol{\theta}}, \alpha_{\boldsymbol{w}}$, soft update parameter $\tau$.
2: **Initialize:**
3:     Replay buffer $\mathcal{D} \leftarrow \varnothing$
4:     Weights snapshots $(\boldsymbol{\theta}_s, \boldsymbol{\theta}_{ss}, \boldsymbol{w}_s, \boldsymbol{w}_{ss}) \leftarrow (\boldsymbol{\theta}, \boldsymbol{\theta}, \boldsymbol{w}, \boldsymbol{w})$
5: **for all** episode $e = 1$ to $T$ **do**
6:     Sample initial state $\mathbf{x}$ from $\mathcal{E}$
7:     $step \leftarrow 1$
8:     **repeat**
9:       **if** $step \leq T_{\mathrm{rand}}$ **then**
10:         Randomly select actions for each agent $i$
11:       **else**
12:         Select actions using policy for each agent $i$
13:       **end if**
14:       Execute actions $\mathbf{a}$, observe rewards $\mathbf{r}$ and new state $\hat{\mathbf{x}}$
15:       Store $(\mathbf{x}, \mathbf{a}, \mathbf{r}, \hat{\mathbf{x}})$ in replay buffer $\mathcal{D}$
16:       $\mathbf{x} \leftarrow \hat{\mathbf{x}}$
17:       **if** $step \% T_{\mathrm{learn}} == 0$ **then**
18:         Sample a batch $B$ from $\mathcal{D}$
19:         Use $B$ and update to solve VI($F_{\mathrm{MADDPG}}, \mathbb{R}^d$) using $\mathcal{B}$
20:         Update target networks:
21:           $\boldsymbol{\theta}' \leftarrow \tau \boldsymbol{\theta} + (1 - \tau) \boldsymbol{\theta}'$
22:           $\boldsymbol{w}' \leftarrow \tau \boldsymbol{w} + (1 - \tau) \boldsymbol{w}'$
23:       **end if**
24:       $step \leftarrow step + 1$
25:     **until** environment terminates
26:     NESTEDLOOKAHEAD($N, e, \boldsymbol{\theta}, \boldsymbol{w}, k_s, k_{ss}, \alpha_{\boldsymbol{\theta}}, \alpha_{\boldsymbol{w}}$)
27: **end for**
28: **Output:** $\boldsymbol{\theta}, \boldsymbol{w}$

---

**LA-MADDPG.** Algorithm 2 describes the *LA-MADDPG* method. Critically, the (nested)LA-VI method is used in the joint strategy space of all players. In this way, the averaging steps address the rotational component of the associated vector field defined by $F_{\mathrm{MADDPG}}$ resulting from the adversarial nature of the agents' objectives. In particular, it is necessary not to use an agent whose parameters have already been averaged at that iteration.

The LA-MADDPG algorithm saves snapshots of the actor and critic networks for all agents, periodically averaging them with the current networks during training. While the MADDPG algorithm (Algorithm 4) runs normally using a base optimizer (e.g., Adam), at every interval $k$, a lookahead averaging step is performed between the current networks (denoted $\boldsymbol{\theta}, \boldsymbol{w}$), and their saved snapshots $\boldsymbol{\theta}_s, \boldsymbol{w}_s$, as detailed in Algorithm 1. This method updates both the current networks and snapshots with the $\alpha$-averaged values. Multiple nested lookahead levels can be applied, where each additional level updates its snapshot after a longer interval; see Algorithm 1. We denote lookahead update intervals (episodes) with $k$ subscripted by $s$ and a larger number of $s$ in subscript implies outer lookahead level, e.g., $k_s, k_{ss}, k_{sss}$ for three levels. All agents undergo lookahead updates at the same step, applying this to both the actor and critic parameters simultaneously. An extended version of the algorithm with more detailed notations can be found in appendix in algorithm 5.

**(LA-)EG-MADDPG.** For **EG-MADDPG**, EG is used for both the actor and critic networks and for all agents; see Algorithm 6 for details. Algorithm 2 can also be used with EG as the base optimizer—an option abstracted by $\mathcal{B}$ in Algorithm 2—resulting in **LA-EG-MADDPG**.

**On the convergence.** Under standard assumptions on the agents' reward functions, such as convexity, the above VI becomes monotone (see Appendix A.1.1 for definition). In this case, the above methods have convergence guarantees, that is EG-MADDPG (Korpelevich, 1976; Gorbunov et al., 2022), LA-MADDPG (Pethick et al., 2023), and LA-EG-MADDPG (Chavdarova et al., 2021, Thm. 3) have convergence guarantees. Contrary to these, the standard gradient descent method does not converge for this problem (Korpelevich, 1976).

## 5 EXPERIMENTS

### 5.1 SETUP

We build upon the open-source *PyTorch* implementation of MADDPG (Lowe et al., 2017)[1]. We use the same hyperparameter settings as specified in the original paper; detailed in Appendix A.2. For our experiments, we use two zero-sum games: the *Rock-Paper-Scissors* (RPS) game and *Matching pennies*. We then apply the methods to two of the *Multi-agent Particle Environments* (MPE) (Lowe et al., 2017). We used versions of the games from the *PettingZoo* (Terry et al., 2021) library. We used five different random seeds for training for all games and trained for 50000 episodes per seed for Matching pennies and 60000 for the rest.

**2-player game: rock–paper–scissors.** Rock–paper–scissors is a widely studied game in multi-agent settings because, in addition to its analytically computable Nash equilibrium that allows for a precise performance measure, it demonstrates interesting cyclical behavior (Zhou, 2015; Wang et al., 2014). The game, with $M = 3$ actions, has a mixed Nash equilibrium where each action is played with equal probability. At equilibrium, each agent's action distribution is $(\frac{1}{3}, \frac{1}{3}, \frac{1}{3})$. This equilibrium allows us to assess the alignment of learned policies with the optimal strategy. In our experiments version, $N$ players compete in an $M$-action game over $t$ steps. At each step, players receive an observation of their opponent's last action. Once all players have selected their actions for the current step, rewards are assigned to each player: as of $-1$ for a losing, 0 for tie and $+1$ for winning the game. we used $N = 2$ players, $M = 3$ actions, and a time horizon of $t = 25$ steps.

**2-player game: matching pennies.** The game has $M = 2$ actions, $N = 2$ players: even and odd that compete over $t$ steps. At each step, the players must choose between two actions: Heads or Tails. Even player wins with a reward of $+1$ if the players chose the same action and loses with a $-1$ otherwise, and vice versa. We used $t = 25$ steps. Similar to Rock–paper–scissors, this game also has mixed Nash equilibrium where each action is played with equal probability. At equilibrium, each agent's action distribution is $(\frac{1}{2}, \frac{1}{2})$.

We measured and plotted the squared norm of the learned policy probabilities relative to the equilibrium for both *rock–paper–scissors* and *matching pennies*.

**MPE: Predator-prey**— from the *Multi-Agent Particle Environments* (MPE) benchmark (Lowe et al., 2017). It consists of $N$ *good* agents, $L$ landmarks, and $M$ *adversary* agents. The good agents are faster and receive negative rewards if caught by adversaries, while the slower adversary agents are rewarded for catching a good agent. All agents can observe the positions of other agents, and adversaries also observe the velocities of the good agents. Additionally, good agents are penalized for going out of bounds. This environment combines elements of both competition and collaboration. While all adversaries are rewarded when one of them catches a good agent, their slower speed typically requires them to collaborate, especially since there are usually more adversaries than good agents. For our experiments, we set $N = 1$, $M = 2$, and $L = 2$.

**MPE: Physical deception**, (Lowe et al., 2017). The game has $N$ good agents, one adversary agent, and $N$ landmarks, with one designated as the target. The adversary does not observe the target and must infer which of the $N$ landmarks is the target one, aiming to get as close as possible and receiving rewards based on its distance from the target. The good agents can observe the target and aim to

---

[1]Available at `https://github.com/Git-123-Hub/maddpg-pettingzoo-pytorch/tree/master`.

deceive the adversary while also staying as close as possible to the target. All good agents share the same reward, based on a combination of their minimum distance to the target and the adversary's distance. This game has no "competitive component" for the adversary: its reward depends solely on its own policy. In our experiments, we set $N = 2$.

**Methods.** We evaluate our proposed methods by comparing them to the baseline, which is the original MADDPG algorithm using Adam (Kingma & Ba, 2015) as the optimizer for all networks. Throughout the section, we will refer to the LA-MADDPG, EG-MADDPG, and LA-EG-MADDPG methods as LA, EG, and LA-EG, respectively. When referring to nLA-based methods, we will indicate the $k$ values for each lookahead level in brackets. For example, LA $(10, 1000)$ represents a two-level lookahead with $k_s = 10$ and $k_{ss} = 1000$. We also use Adam in combination with the VI methods for consistency with the baseline.

Details on the remaining hyperparameters can be found in Appendix A.2.

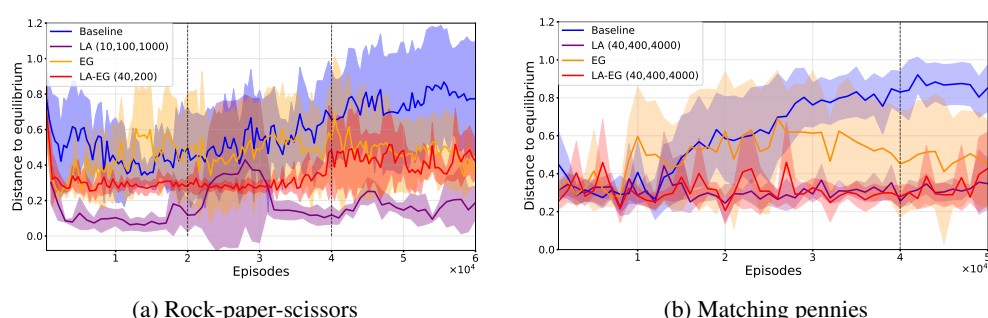

(a) Rock-paper-scissors                                      (b) Matching pennies

Figure 1: **Comparison on the rock–paper–scissors game and matching pennies game between the *GD-MADDPG*, *LA-MADDPG*, *EG-MADDPG* and *LA-EG-MADDPG* methods, denoted as *Baseline*, *LA*, *EG*, *LA-EG*, resp.** $x$-axis: training episodes. $y$-axis: total distance of agents' policies to the equilibrium policy, averaged over 5 seeds. The dotted line depicts the start of the "shifting" (in first-in-first-out order) of the experiences in the buffer.

## 5.2 RESULTS

**2-player games: rock–paper–scissors and matching pennies.** Figures 1a and 1b depict the average distance of the agents' learned policies from the equilibrium policy. The baseline method eventually diverges. In contrast, LA-MADDPG consistently converges to a near optimal policy, outperforming the baseline. While EG-MADDPG behaves similarly to the baseline, combining it with Lookahead stabilizes the performances. Additionally, Adam exhibits high variance across different seeds, while Lookahead significantly reduces variance, providing more stable and reliable results—an important factor in MARL experiments.

For LA, we used $0.5$ for the $\alpha$ hyperparameter, and after experimenting with several values for $k$, we observed that smaller $k$-values for the innermost LA-averaging works better. Refer to Appendix A.2.1

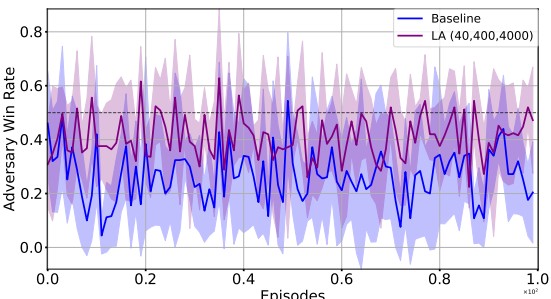

Figure 2: **Comparison on the MPE:Predator-prey game between the *GD-MADDPG* and *LA-MADDPG*, optimization methods, denoted as *Baseline*, *LA*, resp.** $x$-axis: evaluation episodes. $y$-axis: average win rate of adversary agents, averaged over 5 runs with different seeds. The dotted line depicts the desired win rate $(0.5)$ if both agents learn good policies.

| Method | Adversary Win Rate |
|---|---|
| Baseline | $0.45 \pm .16$ |
| LA-MADDPG | $0.53 \pm .11$ |
| EG-MADDPG | $0.56 \pm .27$ |
| LA-EG-MADDPG | $\mathbf{0.51 \pm .14}$ |

Table 1: Means and standard deviations (over 5 seeds) of **adversary win rate on last training episode for MPE: Physical deception**, on 100 test environments. The *win rate* is the fraction of times the adversary was closer to the target. *Closer to* 0.5 *is better.*

for further discussion. Despite relatively little hyperparameter tuning, the results indicate consistent improvement.

**MPE: Predator-prey.** Figure 2 depicts the win rate of the adversary against the good agents. While typical training monitors average rewards to indicate convergence, we observed that after training, one adversary learns to chase the good agent while the other's policy diverges, causing it to move away or wander aimlessly. This suggests a convergence issue in the joint policy space, where one agent's strategy is affected by the other's. Our results in Figure 2 demonstrate that using Algorithm 2 improves this behavior, with both adversaries learning to chase the good agent, reflected in a higher win rate. Full method comparisons are provided in Appendix A.3.3.

**MPE: Physical deception.** Table 1 lists the mean and standard deviation of the adversary's win rate, indicating how often it managed to be closer to the target. Agents reach equilibrium when both teams win with equal probability across multiple instances. Thus, we used 100 test environments per method per seed. Given the game's cooperative nature, the baseline performs relatively well, with EG-MADDPG showing similar performance. Both LA-MADDPG and LA-EG-MADDPG outperform their respective base optimizers–baseline and EG-MADDPG.

**Summary.** Overall, our results indicate the following. *(i)* With hyperparameter tuning, the proposed VI-based methods achieve significant performance improvements; see Figure 1a. *(ii)* With informed guesses for hyperparameters (details in Appendix A.2.1), our VI-based methods consistently outperform the baseline methods. *(iii)* Overall, the proposed VI methods do not yield worse performance than their respective baseline methods, as they effectively address the rotational dynamics.

**Comparison among our VI methods & contrasting with GAN conclusions.** The widely used Extragradient (EG) method for solving VIs—known for its convergence for monotone VIs—overall performs close to the baseline. EG only introduces a minor local adjustment compared to GD. As such, the results align with expectations: while EG occasionally outperforms GD (the baseline), its performance is often similar. In contrast, nested LA applies a significantly stronger contraction, with the degree of contraction increasing as the number of nested levels increases. This leads to substantial performance gains, particularly in terms of stability, as it prevents the last iterate from diverging. However, if the number of nested levels is too high, the steps can become overly conservative or slow. Based on our experiments, three levels of nested LA yielded the best results (see Fig. 1-a). The results also confirm that the MARL vector field in these games is highly rotational. For scenarios with highly competitive reward structures among agents, we recommend using VI methods with higher contractiveness, such as employing multiple levels of nested LA.

These observations are consistent with results from GANs settings (Chavdarova et al., 2021), while EG offers slight improvements over the baseline, more contractive methods consistently achieve better results.

**On the rewards as a metric in MARL.** While saturating rewards are commonly used as a performance metric in MARL, our experiments suggest otherwise, consistent with observations made in some previous works such as (Bowling, 2004). In multi-agent games like Rock-paper-scissors, rewards may converge to a target value even with suboptimal policies, leading to misleading evaluations. For instance, in Figure 3 (top row), agents repeatedly choose similar actions, resulting in ties that yield the correct reward but fail to reach equilibrium—leaving them vulnerable to exploitation by a more skilled opponent. Conversely, LA-MADDPG (bottom row) did not fully converge to the maximum reward, but agents learned near-optimal policies by randomizing over their actions, which is the desired equilibrium. This underscores the need for stronger evaluation metrics in multi-agent reinforcement learning, particularly when the true equilibrium remains unknown. Refer to Appendix A.5 for additional discussion.

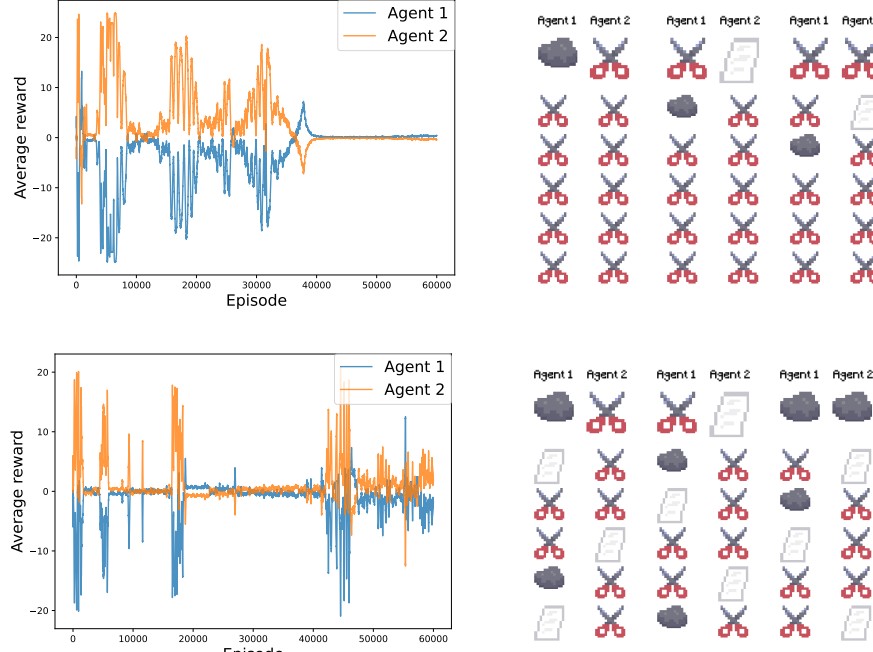

Figure 3: **Saturating rewards (left) versus actions of the learned policies at the end (right) in the rock–paper–scissors game. Top row:** *GD-MADDPG*; **bottom row:** *LA-MADDPG*. In the left column, blue and orange show the running average of rewards through a window of 100 episodes. In the right column, we depict actions from the respective learned policies evaluation after training is completed, where each row represents what actions players have chosen in one step of the episode. Saturating rewards do not imply good performance, as evidenced by the top row; refer to Section 5.2 for discussion.

## 6 CONCLUSION

This paper addresses the fundamental optimization challenges in multi-agent reinforcement learning (MARL). By framing MARL as an instance of a Variational Inequality (VI) problem, we highlight its inherent optimization difficulties, which resemble those encountered in solving VIs. These challenges manifest in practice as notoriously difficult training, significant performance variability across random seeds, and other issues that hinder MARL reproducibility, development, and deployment.

To address these challenges, we leverage VI optimization techniques to enhance the convergence and stability of MARL methods. We introduced the *LA-MADDPG*, *EG-MADDPG* and *LA-EG-MADDPG* algorithms that combine the multi-agent deep deterministic policy gradient (MADDPG) method with nested Lookahead-VI (Chavdarova et al., 2021), Extragradient (Korpelevich, 1976), and a combination of both, respectively. Our experiments on the *rock-paper-scissors, matching pennies* and two *MPE* environments (Lowe et al., 2017) consistently demonstrated the effectiveness of the VI variants of MADDPG in improving performance and stabilizing training compared to the standard baseline method. These findings point toward promising opportunities for further development of VI-based methods in MARL, particularly in leveraging the structure of the MARL optimization landscape.

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

# A APPENDIX

## A.1 ADDITIONAL BACKGROUND

### A.1.1 VI CLASSES AND ADDITIONAL METHODS

The following VI class is often referred to as the generalized class for VIs to that of convexity in minimization.

**Definition 1 (monotonicity)** *An operator* $F : \mathbb{R}^d \to \mathbb{R}^d$ *is* monotone *if* $\langle z - z', F(z) - F(z') \rangle \geq 0, \ \forall z, z' \in \mathbb{R}^d$. *$F$ is $\mu$-strongly monotone if:* $\langle z - z', F(z) - F(z') \rangle \geq \mu \|z - z'\|^2$ *for all* $z, z' \in \mathbb{R}^d$.

In addition to those presented in the main part, we describe the following popular VI method.

**Optimistic Gradient Descent (OGD).** The update rule of Optimistic Gradient Descent OGD ((OGD) Popov, 1980) is:

$$z_{t+1} = z_t - 2\eta F(z_t) + \eta F(z_{t-1}), \tag{OGD}$$

where $\eta \in (0, 1)$ is the learning rate.

### A.1.2 PSEUDOCODE FOR NESTED LOOKAHEAD FOR A TWO-PLAYER GAME

For completeness, in Algorithm 3 we give the details of the nested Lookahead-Minmax algorithm proposed in (Algorithm 6, Chavdarova et al., 2021) with two-levels.

---

**Algorithm 3** Pseudocode of Two-Level Nested Lookahead–Minmax. (Chavdarova et al., 2021)

---
1: **Input:** Stopping time $T$, learning rates $\eta_\theta, \eta_\varphi$, initial weights $\theta, \varphi$, lookahead hyperparameters $k_s, k_{ss}$ and $\alpha$, losses $\mathcal{L}^\theta, \mathcal{L}^\varphi$, update ratio $r$, real–data distribution $p_d$, noise–data distribution $p_z$.

2: $(\theta_s, \theta_{ss}, \varphi_s, \varphi_{ss}) \leftarrow (\theta, \theta, \varphi, \varphi)$        *(store copies for slow and super-slow)*
3: **for** $t \in 1, \ldots, T$ **do**
4:  **for** $i \in 1, \ldots, r$ **do**
5:   $x \sim p_d, z \sim p_z$
6:   $\varphi \leftarrow \varphi - \eta_\varphi \nabla_\varphi \mathcal{L}^\varphi(\theta, \varphi, x, z)$         *(update $\varphi$ r times)*
7:  **end for**
8:  $z \sim p_z$
9:  $\theta \leftarrow \theta - \eta_\theta \nabla_\theta \mathcal{L}^\theta(\theta, \varphi, z)$          *(update $\theta$ once)*
10:  **if** $t \% k_s == 0$ **then**
11:   $\varphi \leftarrow \varphi_s + \alpha_\varphi(\varphi - \varphi_s)$     *(backtracking on interpolated line $\varphi_s, \varphi$)*
12:   $\theta \leftarrow \theta_s + \alpha_\theta(\theta - \theta_s)$     *(backtracking on interpolated line $\theta_s, \theta$)*
13:   $(\theta_s, \varphi_s) \leftarrow (\theta, \varphi)$        *(update slow checkpoints)*
14:  **end if**
15:  **if** $t \% k_{ss} == 0$ **then**
16:   $\varphi \leftarrow \varphi_{ss} + \alpha_\varphi(\varphi - \varphi_{ss})$    *(backtracking on interpolated line $\varphi_{ss}, \varphi$)*
17:   $\theta \leftarrow \theta_{ss} + \alpha_\theta(\theta - \theta_{ss})$    *(backtracking on interpolated line $\theta_{ss}, \theta$)*
18:   $(\theta_{ss}, \varphi_{ss}) \leftarrow (\theta, \varphi)$      *(update super-slow checkpoints)*
19:   $(\theta_s, \varphi_s) \leftarrow (\theta, \varphi)$       *(update slow checkpoints)*
20:  **end if**
21: **end for**
22: **Output:** $\theta_{ss}, \varphi_{ss}$

---

### A.1.3 DETAILS ON THE MADDPG ALGORITHM

The MADDPG algorithm is outlined in Algorithm 4. An empty replay buffer $\mathcal{D}$ is initialized to store experiences (line 3). In each episode, the environment is reset and experiences in the form of *(state, action, reward, next state)* are saved to $\mathcal{D}$. After a predetermined number of random iterations, learning begins by sampling batches from $\mathcal{D}$.

The critic of agent $i$ receives the sampled joint actions $\boldsymbol{a}$ of all agents and the state information of agent $i$ to output the predicted $Q_i$-value of agent $i$. Deep Q-learning (Mnih et al., 2015) is then used to update the critic network; lines 21-22. Then, the agents' policy network is optimized using policy gradient; refer to 24. Finally, following each learning iteration, the target networks are updated towards current actor and critic networks using a fraction $\tau$.

All networks are optimized using the Adam optimizer (Kingma & Ba, 2015). Once training is complete, each agent's actor operates independently during execution. This approach is applicable across cooperative, competitive, and mixed environments.

---

**Algorithm 4** Pseudocode for MADDPG (Lowe et al., 2017).

---

1: **Input:** Environment $\mathcal{E}$, number of agents $N$, number of episodes $T$, action spaces $\{\mathcal{A}_i\}_{i=1}^N$, number of random steps $T_{\text{rand}}$ before learning, learning interval $T_{\text{learn}}$, actor networks $\{\boldsymbol{\mu}_i\}_{i=1}^N$, with initial weights $\boldsymbol{\theta} \equiv \{\boldsymbol{\theta}_i\}_{i=1}^N$, critic networks $\{\mathbf{Q}_i\}_{i=1}^N$ with initial weights $\boldsymbol{w} \equiv \{\boldsymbol{w}_i\}_{i=1}^N$, learning rates $\eta_{\boldsymbol{\theta}}, \eta_{\boldsymbol{w}}$, optimizer $\mathcal{B}$ (e.g., Adam), discount factor $\gamma$, soft update parameter $\tau$.

2: **Initialize:**

3:     Replay buffer $\mathcal{D} \leftarrow \varnothing$

4: **for all** episode $e \in 1, \ldots, T$ **do**

5:     $\mathbf{x} \leftarrow Sample(\mathcal{E})$          *(sample from environment $\mathcal{E}$)*

6:     $step \leftarrow 1$

7:     **repeat**

8:       **if** $e \leq T_{\text{rand}}$ **then**

9:         for each agent $i$, $a_i \sim \mathcal{A}_i$          *(sample actions randomly)*

10:       **else**

11:         for each agent $i$, select action $a_i = \boldsymbol{\mu}_i(o_i) + \mathcal{N}_t$ using current policy and exploration

12:       **end if**

13:                             *(apply actions and record results)*

14:       Execute actions $\mathbf{a} = (a_1, \ldots, a_N)$, observe rewards $\mathbf{r}$ and new state $\hat{\mathbf{x}}$

15:       replay buffer $\mathcal{D} \leftarrow (\mathbf{x}, \mathbf{a}, \mathbf{r}, \hat{\mathbf{x}})$

16:       $\mathbf{x} \leftarrow \hat{\mathbf{x}}$

17:                             *(apply learning step if applicable)*

18:       **if** $step \% T_{\text{learn}} = 0$ **then**

19:         **for all** agent $i \in 1, \ldots, N$ **do**

20:           sample batch $\{(\mathbf{x}^j, \mathbf{a}^j, \mathbf{r}^j, \hat{\mathbf{x}}^j)\}_{j=1}^B$ of size $B$ from $\mathcal{D}$

21:           $y^j \leftarrow r_i^j + \gamma \mathbf{Q}^{\boldsymbol{\mu}'}(\hat{\mathbf{x}}^j, a_1', \ldots, a_N')$, where $\mathbf{a}_k' = \{\boldsymbol{\mu}_k'(o_k^j)\}$

22:           Update critic by minimizing the loss (using optimizer $\mathcal{B}$ ):

$$\mathcal{L}(\boldsymbol{\theta}_i) = \tfrac{1}{S} \sum_j \left( y^j - \mathbf{Q}_i^{\boldsymbol{\mu}}(\mathbf{x}^j, a_1^j, \ldots, a_N^j) \right)^2$$

23:           Update actor policy using policy gradient formula and optimizer $\mathcal{B}$

24:           $\nabla_{\boldsymbol{\theta}_i} J \approx \tfrac{1}{S} \sum_j \nabla_{\boldsymbol{\theta}_i} \boldsymbol{\mu}_i(o_i^j) \nabla_{a_i} \mathbf{Q}_i^{\boldsymbol{\mu}}(\mathbf{x}^j, a_1^j, \ldots, a_i, \ldots, a_N^j)$, where $a_i = \boldsymbol{\mu}_i(o_i^j)$

25:         **end for**

26:         **for all** agent $i \in 1, \ldots, N$ **do**

27:           $\boldsymbol{\theta}_i' \leftarrow \tau \boldsymbol{\theta}_i + (1 - \tau) \boldsymbol{\theta}_i'$          *(update target networks)*

28:           $\boldsymbol{w}_i' \leftarrow \tau \boldsymbol{w}_i + (1 - \tau) \boldsymbol{w}_i'$

29:         **end for**

30:       **end if**

31:       $step \leftarrow step + 1$

32:     **until** environment terminates

33: **end for**

34: **Output:** $\boldsymbol{\theta}, \boldsymbol{w}$

---

### A.1.4 EXTENDED VERSION OF LA-MADDPG PSEUDOCODE

We include an extended version for the LA-MADDPG algorithm without VI notations in algorithm 5.

**Algorithm 5** Pseudocode for LA–MADDPG: MADDPG with (Nested) Lookahead.

1: **Input:** Environment $\mathcal{E}$, number of agents $N$, number of episodes $T$, action spaces $\{\mathcal{A}_i\}_{i=1}^N$, number of random steps $T_{\text{rand}}$ before learning, learning interval $T_{\text{learn}}$, actor networks $\{\boldsymbol{\mu}_i\}_{i=1}^N$, with initial weights $\boldsymbol{\theta} \equiv \{\boldsymbol{\theta}_i\}_{i=1}^N$, critic networks $\{\mathbf{Q}_i\}_{i=1}^N$ with initial weights $\boldsymbol{w} \equiv \{\boldsymbol{w}_i\}_{i=1}^N$, learning rates $\eta_{\boldsymbol{\theta}}, \eta_{\boldsymbol{w}}$, base optimizer $\mathcal{B}$ (e.g., Adam), discount factor $\gamma$, lookahead hyperparameters $k_s, k_{ss}$ (where $k_{ss}$ can be $\varnothing$) and $\alpha_{\boldsymbol{\theta}}, \alpha_{\boldsymbol{w}}$, soft update parameter $\tau$.
2: **Initialize:**
3:     Replay buffer $\mathcal{D} \leftarrow \varnothing$
4:     **for all** agent $i \in 1, \ldots, N$ **do**
5:         $(\boldsymbol{\theta}_{i,s}, \boldsymbol{\theta}_{i,ss}, \boldsymbol{w}_{i,s}, \boldsymbol{w}_{i,ss}) \leftarrow (\boldsymbol{\theta}_i, \boldsymbol{\theta}_i, \boldsymbol{\theta}_i, \boldsymbol{w}_i, \boldsymbol{w}_i, \boldsymbol{w}_i)$
6:                                                             *(store snapshots for nLA)*
7:     **end for**
8: **for all** episode $e \in 1, \ldots, T$ **do**
9:     $\mathbf{x} \leftarrow Sample(\mathcal{E})$                                     *(sample from environment $\mathcal{E}$)*
10:     $step \leftarrow 1$
11:     **repeat**
12:         **if** $e \leq T_{\text{rand}}$ **then**
13:             for each agent $i$, $a_i \sim \mathcal{A}_i$                      *(sample actions randomly)*
14:         **else**
15:             for each agent $i$, select action $a_i$ using current policy and exploration
16:         **end if**
17:                                      *(apply actions and record results)*
18:         Execute actions $\mathbf{a} = (a_1, \ldots, a_N)$, observe rewards $\mathbf{r}$ and new state $\hat{\mathbf{x}}$
19:         replay buffer $\mathcal{D} \leftarrow (\mathbf{x}, \mathbf{a}, \mathbf{r}, \hat{\mathbf{x}})$
20:         $\mathbf{x} \leftarrow \hat{\mathbf{x}}$
21:                                     *(apply learning step if applicable)*
22:         **if** $step \% T_{\text{learn}} = 0$ **then**
23:             **for all** agents $i \in 1, \ldots, N$ **do**
24:                 sample batch $\{(\mathbf{x}^j, \mathbf{a}^j, \mathbf{r}^j, \hat{\mathbf{x}}^j)\}_{j=1}^B$ of size $B$ from $\mathcal{D}$
25:                 $y^j \leftarrow r_i^j + \gamma \mathbf{Q}^{\boldsymbol{\mu}'}(\hat{\mathbf{x}}^j, a_1', \ldots, a_N')$, where $\mathbf{a}_k' = \{\boldsymbol{\mu}_k'(o_k^j)\}$
26:                 Update critic by minimizing the loss $\mathcal{L}(\boldsymbol{w}_i) = \frac{1}{S} \sum_j \left( y^j - \mathbf{Q}_i^{\boldsymbol{\mu}}(\mathbf{x}^j, a_1^j, \ldots, a_N^j) \right)^2$ using $\mathcal{B}$
27:                 Update actor policy using policy gradient formula and $\mathcal{B}$
28:                 $\nabla_{\boldsymbol{\theta}_i} J \approx \frac{1}{S} \sum_j \nabla_{\boldsymbol{\theta}_i} \boldsymbol{\mu}_i(o_i^j) \nabla_{a_i} \mathbf{Q}_i^{\boldsymbol{\mu}}(\mathbf{x}^j, a_1^j, \ldots, a_i, \ldots, a_N^j)$, where $a_i = \boldsymbol{\mu}_i(o_i^j)$
29:             **end for**
30:             **for all** agents $i \in 1, \ldots, N$ **do**
31:                 $\boldsymbol{\theta}_i' \leftarrow \tau \boldsymbol{\theta}_i + (1 - \tau) \boldsymbol{\theta}_i'$                    *(update target networks)*
32:                 $\boldsymbol{w}_i' \leftarrow \tau \boldsymbol{w}_i + (1 - \tau) \boldsymbol{w}_i'$
33:             **end for**
34:         **end if**
35:         $step \leftarrow step + 1$
36:     **until** environment terminates
37:     NESTEDLOOKAHEAD$(N, e, \boldsymbol{\Theta}, \mathbf{W}, k_s, k_{ss}, \alpha_{\boldsymbol{\theta}}, \alpha_{\boldsymbol{w}})$
38:         where:
39:         $\boldsymbol{\Theta} = \{(\boldsymbol{\theta}_i, \boldsymbol{\theta}_{i,s}, \boldsymbol{\theta}_{i,ss}\}_{i=1}^N$                *(all actor weights and snapshots)*
40:         $\mathbf{W} = \{(\boldsymbol{w}_i, \boldsymbol{w}_{i,s}, \boldsymbol{w}_{i,ss})\}_{i=1}^N$            *(all critic weights and snapshots)*
41: **end for**
42: **Output:** $\boldsymbol{\theta}, \boldsymbol{w}$

### A.1.5 Pseudocode for Extragradient

In Algorithm 6 outlines the *Extragradient* optimizer (Korpelevich, 1976), which we employ in EG-MADDPG. This method uses a gradient-based optimizer to compute the extrapolation iterate, then applies the gradient at the extrapolated point to perform an actual update step. The extragradient optimizer is used to update all agents' actor and critic networks. In our experiments, we use Adam for both the extrapolation and update steps, maintaining the same learning intervals and parameters as in the baseline algorithm.

---

**Algorithm 6** Extragradient optimizer; Can be used as $\mathcal{B}$ in algorithm 2.

---

1: **Input:** learning rate $\eta_{\varphi}$, initial weights $\varphi$, loss $\mathcal{L}^{\varphi}$, extrapolation steps $t$
2: $\varphi^{copy} \leftarrow \varphi$                                                 *(Save current parameters)*
3: **for** $i \in 1, \ldots, t$ **do**
4:     $\varphi = \varphi - \eta_{\varphi}\nabla_{\varphi}\mathcal{L}^{\varphi}(\varphi)$                           *(Compute the extrapolated $\varphi$)*
5: **end for**
6: $\varphi = \varphi^{copy} - \eta_{\varphi}\nabla_{\varphi}\mathcal{L}^{\varphi}(\varphi)$                               *(update $\varphi$)*
7: **Output:** $\varphi$

---

## A.2 Details On The Implementation

As mentioned earlier, we followed the configurations and hyperparameters from the original MAD-DPG paper for our implementation. For completeness, these are listed in Table 2. We ran $T = 60000$ for all environments except Matching Pennies where we ran for $50000$ training episodes, with a maximum of $25$ environment steps ($s$) per episode.

In all Rock-Paper-Scissors and Matching pennies experiments, we used a 2-layer MLP with $64$ units per layer, while for MPE: Predator-prey , we used a 2-layer MLP with $128$ units per layer. ReLU activation was applied between layers for both the policy and value networks of all agents.

### A.2.1 Hyperparameter Selection for Nested-Lookahead

In this section, we discuss and share guidelines for hyperparameter selection based on our experiments.

**Summary.**

- We observed two- or three-level of nested-Lookahead outperform single-level Lookahead.
- Each level has different $k$, denoted here with $k_s, k_{ss}, k_{sss}$ as in the main part. These should be selected as multiple of the selected $k$ for the level before, that is, $k_{ss} \equiv c_{ss}k_s$, and $k_{sss} \equiv c_{sss}k_{ss}$, where $c_{ss}, c_{sss}$ are positive integers.
- We observed that for the innermost lookahead, small values for $k_s$, such as smaller than $50$, perform better than using large values. For the outer $k_{ss}, k_{sss}$ large values work well, such as in the range between $5 - 10$ for the $c_{ss}, c_{sss}$.
- We typically used $\alpha = 0.5$, and we observed lower values, such as $\alpha = 0.3$, give better performances then $\alpha > 0.5$.

**Discussion.**

- To give an intuition regarding the above-listed conclusions, small values for $k_s$ help because the MARL setting is very noisy and the vector field is rotational. If large values are used for $k_s$, then the algorithm will diverge away. It is known that the combination of noise and rotational vector field can cause methods to diverge away (Chavdarova et al., 2019).
- Relative to the analogous conclusions for GANs (Chavdarova et al., 2021), the differences is that:
    - The better-performing values for $k_s$ are of a similar range as for Lookahead with GD for GANs; however they are smaller than those used for Lookahead with EG for GANs.

Table 2: Hyperparameters used for LA-MADDPG experiments.

| Name | Description |
| --- | --- |
| Adam $lr$ | 0.01 |
| Adam $\beta_1$ | 0.9 |
| Adam $\beta_2$ | 0.999 |
| Batch-size | 1024 |
| Update ratio $\tau$ | 0.01 |
| Discount factor $\gamma$ | 0.95 |
| Replay Buffer | $10^6$ |
| learning step $T_{\text{learn}}$ | 100 |
| $T_{\text{rand}}$ | 1024 |
| Lookahead $\alpha$ | 0.5 |

### A.3 ADDITIONAL RESULTS

#### A.3.1 ROCK-PAPER-SCISSORS: BUFFER STRUCTURE

For the Rock-Paper-Scissors (RPS) game, using a buffer size of 1M wasn't sufficient to store all experiences from the 60K training episodes. We observed a change in algorithm behavior around 40K episodes. To explore the impact of buffer configurations, we experimented with different sizes and structures, as experience storage plays a critical role in multi-agent reinforcement learning.

**Full buffer.** The buffer is configured to store all experiences from the beginning to the end of training without any loss.

**Buffer clearing.** In this setup, a smaller buffer is used, and once full, the buffer is cleared completely, and new experiences are stored from the start.

**Buffer shifting.** Similar to the small buffer setup, but once full, old experiences are replaced by new ones in a first-in-first-out (FIFO) manner.

**Results.** Figure 4 depicts the results when using different buffer options for the RPS game.

#### A.3.2 ROCK-PAPER-SCISSORS: SCHEDULED LEARNING RATE

We experimented with gradually decreasing the learning rate (LR) during training to see if it would aid convergence to the optimal policy in RPS. While this approach reduced noise in the results, it also led to increased variance across all methods except for LA-MADDPG.

Figure 5 depicts the average distance to the equilibrium policy over 5 different seeds for each methods, using periodically decreased step sizes.

#### A.3.3 MPE: PREDATOR-PREY FULL RESULTS

While in the main part in Figure 2 we showed only two methods for clarity, Figure 6 depicts all methods.

We also evaluated the trained models of all methods on an instance of the environment that runs for 50 steps to compare learned policies. We present snapshots from it in Figure 7. Here, you can clearly anticipate the difference between the policies from baseline and our optimization methods. As in the baseline, only one agent will chase at the beginning of episode. Moreover, for the baseline (topmost row), the agents move further away from the landmarks and the good agent, which is suboptimal. This can be noticed from the decreasing agents' size in the figures. While in ours, both adversary agents engage in chasing the good agent until the end.

#### A.3.4 MPE: PREDATOR-PREY AND PHYSICAL DECEPTION TRAINING FIGURES

In figures 8a and 8b we include the rewards achieved during the training of GD-MADDPG and LA-MADDPG resp. for MPE: Predator-prey. The figures show individual rewards for the agent

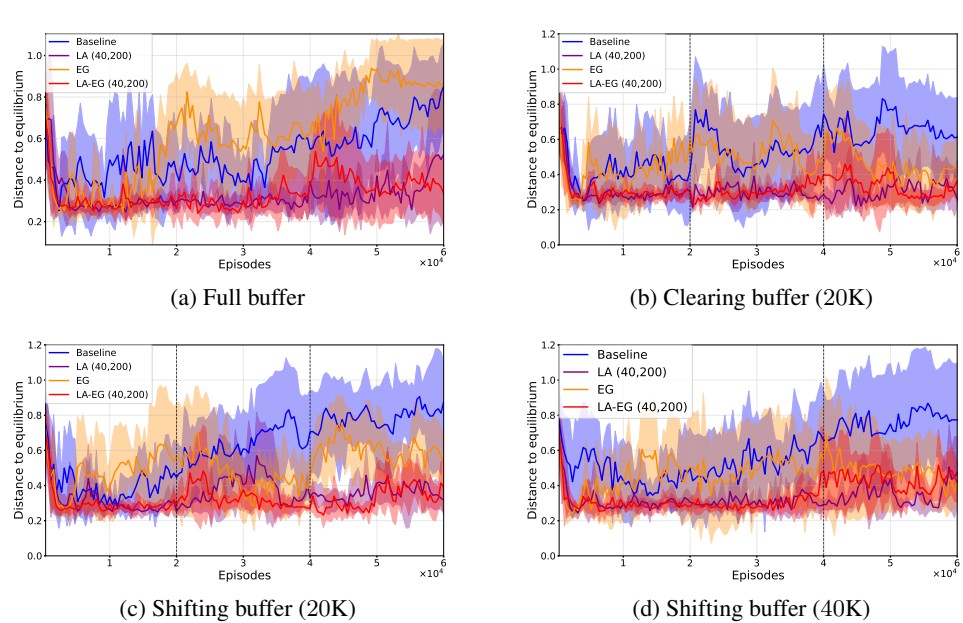

(a) Full buffer

(b) Clearing buffer (20K)

(c) Shifting buffer (20K)

(d) Shifting buffer (40K)

Figure 4: **Comparison of different buffer configurations (see Appendix A.3.1) and methods on Rock-paper-scissors game.** $x$-axis: training episodes. $y$-axis: 5-seed average norm between the two players' policies and equilibrium policy $(\frac{1}{3}, \frac{1}{3}, \frac{1}{3})^2$. The dotted line indicates the point at which the buffer begins to change, either through shifting or clearing.

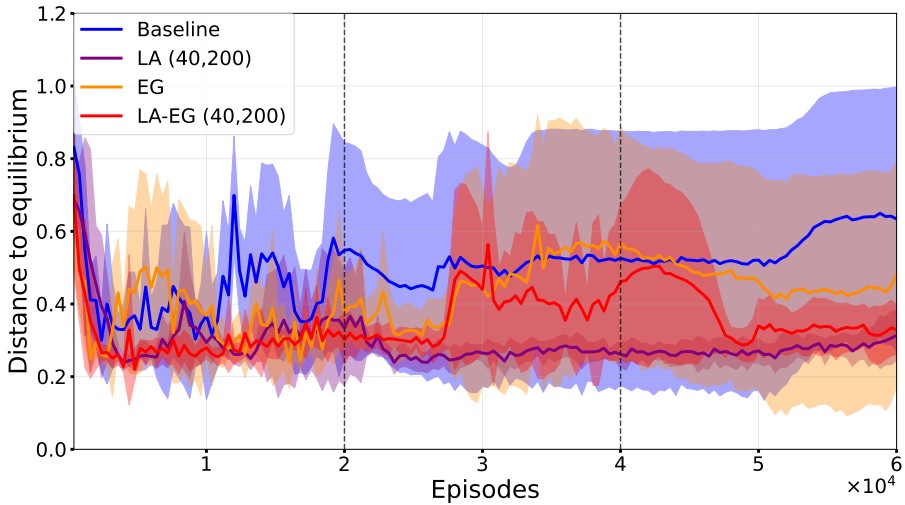

Figure 5: **Compares MADDPG with different *LA-MADDPG* configurations to the baseline MADDPG with (*Adam*) in rock–paper–scissors**. $x$-axis: training episodes. $y$-axis: 5-seed average norm between the two players' policies and equilibrium policy $(\frac{1}{3}, \frac{1}{3}, \frac{1}{3})^2$. The dotted lines depict the times when the learning rate was decreased by a factor of 10.

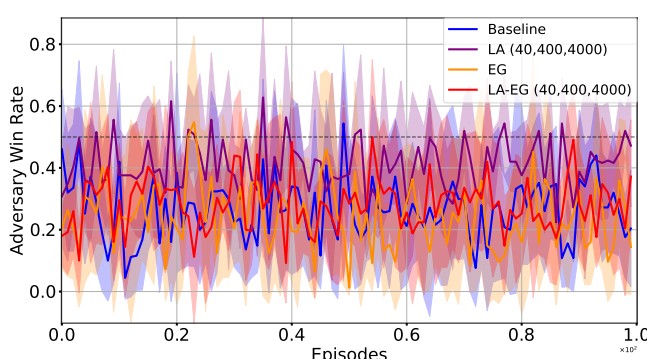

Figure 6: **Comparison on the MPE–Predator-prey game between the** *GD-MADDPG*, *LA-MADDPG*, *EG-MADDPG* **and** *LA-EG-MADDPG* **optimization methods, denoted as** *Baseline*, *LA*, *EG*, *LA-EG*, **resp.** $x$-axis: evaluation episodes. $y$-axis: mean adversaries win rate, averaged over 5 runs with different seeds.

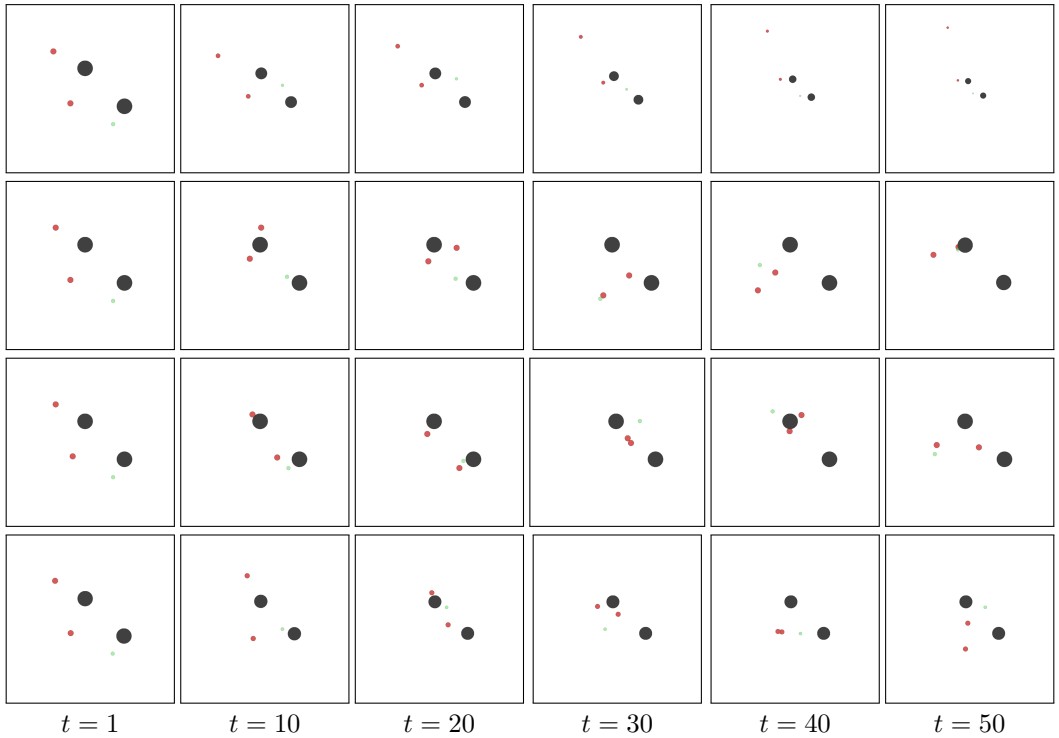

Figure 7: **Agents' trajectories of fully trained models with all considered optimization methods on the same environment seed of MPE: Predator-prey.** Snapshots show the progress of agents as time progresses in a 50 steps long environment. Each row contains snapshots of one method, from top to bottom: *GD-MADDPG*, *LA-MADDPG*, *EG-MADDPG* and *LA-EG-MADDPG*. Big dark circles represent landmarks, small red circles are adversary agents and green one is the good agent.

.

(prey) and one adversary (predator). Blue and green show the individual rewards received at each episode while the orange and red lines are the respective running averages with window size of 100 of those rewards.

Figures 9a and 9b demonstrate same results but for MPE: Physical deception. In this game, We have two good agents, 'Agent 0 and 1' but since they are both receive same rewards, we only show agent 0.

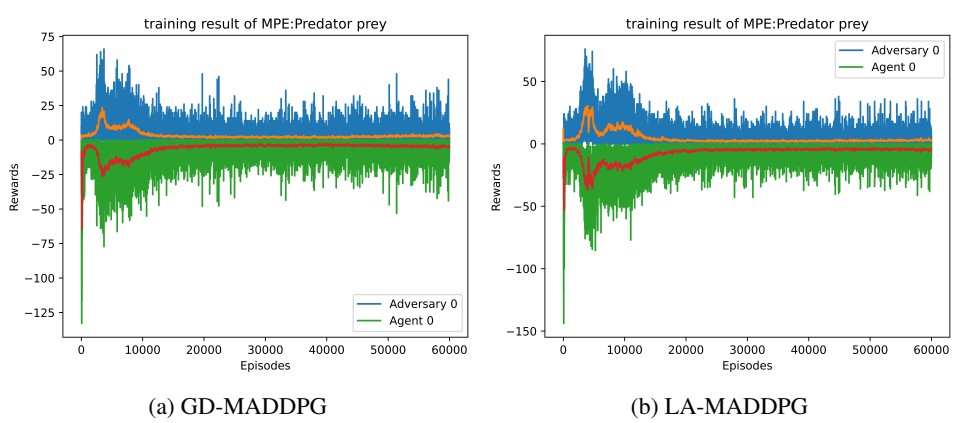

(a) GD-MADDPG         (b) LA-MADDPG

Figure 8: **The figure shows the learning curves during training of GD-MADDPG and LA-MADDPG for MPE: Predator-Prey.** $x$-axis: training episodes. $y$-axis: agents rewards and their moving average with a window size of 100, calculated over 5-seeds over 5 seeds.

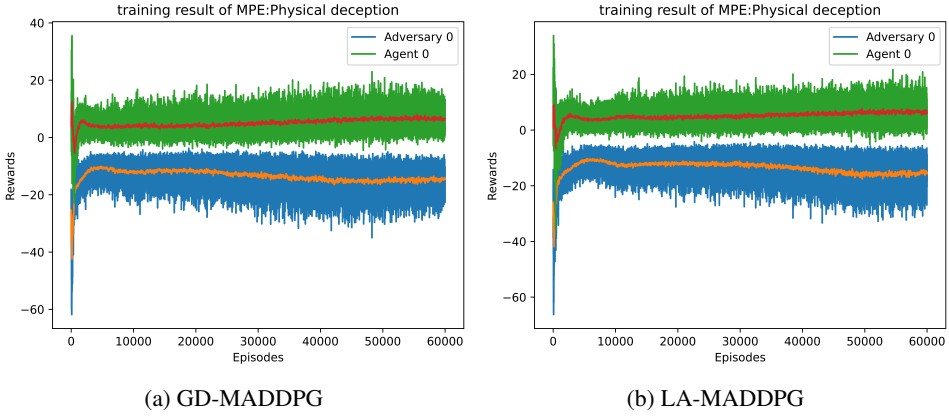

(a) GD-MADDPG         (b) LA-MADDPG

Figure 9: **The figure shows the learning curves during training of GD-MADDPG and LA-MADDPG for MPE: Physical deception.** $x$-axis: training episodes. $y$-axis: agents rewards and their moving average with a window size of 100, calculated over 5-seeds over 5 seeds.

## A.4 MATD3 EXPERIMENTS

We compared Multi-agent TD3 (MATD3), (Ackermann et al., 2019) with MADDPG on the MPE benchmark, *Physical deception*. From Figure 10, we observe that the performances are similar: both algorithms fluctuate between high and low rewards. Hence, optimization methods dealing with rotational dynamics would benefit both.

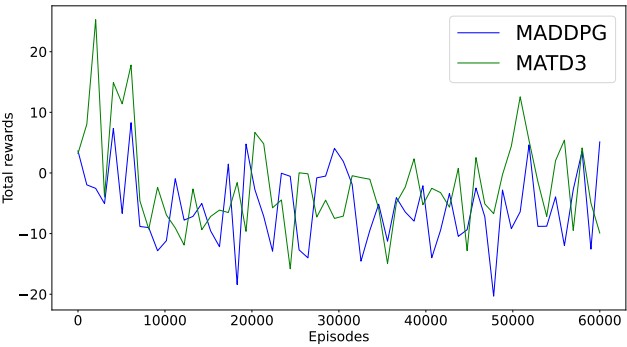

Figure 10: **Comparison on the MPE–Physical deception game between the *MADDPG*, and *MATD3* algorithms.** $x$-axis: training episodes. $y$-axis: total rewards.

## A.5 ON THE REWARDS AS CONVERGENCE METRIC

Based on our experiments and findings from the multi-agent literature (Bowling, 2004), we observe that average rewards offer a weaker measure of convergence compared to policy convergence in multi-agent games. This implies that rewards can reach a target value even when the underlying policy is suboptimal. For example, in the Rock–paper–scissors game, the Nash equilibrium policy leads to nearly equal wins for both players, resulting in a total reward of zero. However, this same reward can also be achieved if one player always wins while the other consistently loses, or if both players repeatedly select the same action, leading to a tie. As such, relying solely on rewards during training can be misleading.

Figure 3 (top row) depicts a case with the baseline where, despite rewards converging during training, the agents ultimately learned to play the same action repeatedly, resulting in ties. Although this matched the expected reward, it falls far short of equilibrium and leaves the agents vulnerable to exploitation by more skilled opponents. In contrast, the same figure shows results from LA-MADDPG under the same experimental conditions. Notably, while the rewards did not fully converge, the agents learned a near-optimal policy during evaluation, alternating between all three actions as expected. These results also align with the findings shown in Figure 1a.

We explored the use of gradient norms as a potential metric in these scenarios but found them to be of limited utility, as they provided no clear indication of convergence for either method. We include those results in Figure 11, where we compare the gradient norms of Adam and LA across the networks of different players.

This work highlights the need for more robust evaluation metrics in multi-agent reinforcement learning, a point also emphasized in (Lanctot et al., 2023), as reward-based metrics alone may be inadequate, particularly in situations where the true equilibrium is unknown.

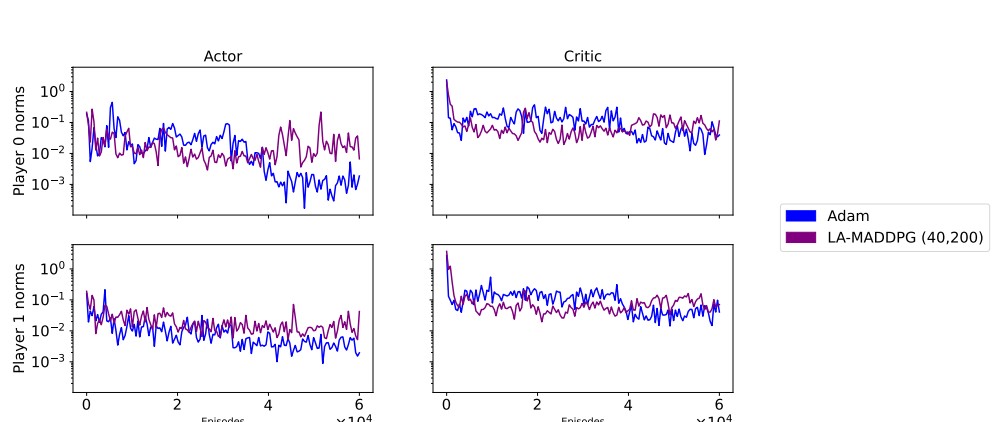

Figure 11: **Gradient norms across training in the *rock–paper–scissors* game.**

