# OpenReview forum: "Variational Inequality Methods for Multi-Agent Reinforcement Learning: Performance and Stability Gains"
_ICLR.cc/2025/Conference — Submitted to ICLR 2025_

### Official Review · Reviewer_UjtH · 2024-10-25

**Soundness:** 3
**Presentation:** 3
**Contribution:** 2
**Rating:** 5
**Confidence:** 3

**Summary:**

This paper proposes a novel optimization method for Multi-Agent RL (MARL), Lookahead-VI, which is a Variational Inequalities(VI)-based methods to substitute Adam optimizer with gradient descent to make MARL algorithms (MADDPG in this paper) better converge. The authors remodel MARL problems as a special case of VI problems, and apply Lookahead-VI to address it. On several simple environments, the proposed method achieves higher performance and can retrieve policies closer to equilibrium.

**Strengths:**

1. The idea to view multi-agent RL as a VI problem and introduce new solvers than gradient descent with Adam is novel and looks mathematically promising, since traditional RL can be seen as a special case of the proposed framework.

2. The paper clearly shows how MADDPG can be reformulated into a VI problem and how gradient descent can be seen as a special case for solving such VI problems, which gives people outside of VI community a good grasp of how the algorithm works.

**Weaknesses:**

1. The contribution to the MARL community is limited by the fact that MADDPG is largely outdated due to its difficulty to train and instability. There are many different, newer methods available, such as MATD3 [1], MAAC [2], MAPPO (which is proposed for cooperative tasks but still applicable to competitive ones; see https://marllib.readthedocs.io/en/latest/algorithm/ppo_family.html#mappo-ppo-agent-with-a-centralized-critic) [3], etc. Many of them are not referenced in this paper. In fact, instability can still be witnessed in Fig. 3 and I am curious whether changing an RL algorithm would fix such issue.

2. While this paper empirically demonstrates that the proposed method is closer to equilibrium, those works are mostly on simple environments. It is unclear whether the property of being closer to equilibrium in simple tasks will bring better performance for more complicated mainstream MARL benchmarks such as SMAC.

**Minor Issues**

1. in line 150-151, the authors wrote "Multi-agent deep deterministic policy gradient (MADDPG, Lowe et al., 2017), extends Deep deterministic policy gradient (DDPG, Lillicrap et al., 2019)". This is not proper reference as an older paper seemingly extends a much newer paper.

2. each $i=1,\dots,N$ agent -> each agent $i=1,\dots,N$ in line 224.

3. The first paragraph of introduction mentions many application of MARL without reference; it would be better if the authors can provide reference for each of the application listed.

**References**

[1] J. Ackermann et al. Reducing Overestimation Bias in Multi-Agent Domains Using Double Centralized Critics. In Deep RL Workshop at NeurIPS, 2019.

[2] S. Iqbal and F. Sha. Actor-Attention-Critic for Multi-Agent Reinforcement Learning. In ICML, 2019.

[3] C. Yu et al. The Surprising Effectiveness of PPO in Cooperative, Multi-Agent Games. In NeurIPS Datasets and Benchmarks, 2022.

**Questions:**

I have several questions:

1. while it is well-known that MARL with gradient descent is hard to converge, is there any theoretical proof that other VI methods are better than gradient descent when it comes to this special case of VI problems?

2. Can the authors explain why EG does not seem to improve performance as significant as LA, and EG-LA-MADDPG does not work well in the MPE-Predator-prey game and the rock-paper-scissor game in Fig. 5?

3. The authors claim "We did not achieve full convergence to the Nash equilibrium with any of the algorithms, as we did not extensively tune the hyperparameters." in line 372-373. Is it possible to reach Nash equilibrium by tuning the hyperparameters more? It would be great if the authors could provide more attempts on achieving Nash equilibrium by tuning hyperparameters (which is also an ablation to hyperparameter sensitivity).

4. About "on the rewards as a metric in MARL": the authors claim that we need stronger evaluation metrics in MARL. While I agree that reward cannot fully reflect agent's behavior, I am not sure if other metrics could be called as "stronger" as the goal of multi-agent RL, like any RL, is to maximize reward.

---

> ### Author Response · Authors · 2024-11-25
>
> Thanks a lot for your time reviewing our paper and for your valuable feedback.
>
>
> ## W1: contribution to the MARL community
>
>
> > MADDPG is largely outdated due to its difficulty to train and instability.
>
> We compared MADDPG with MATD3 ([1] Ackermann et al., 2019); please see Figure $10$ in Appendix A.4 for details. Our observations show that the performance of both algorithms fluctuates between high and low rewards, indicating similar behavior. This suggests that our proposed methods, which address rotational dynamics, would be beneficial for both algorithms. Our experiments are designed to demonstrate relative improvements over the baseline. Therefore, the insights gained from our work are applicable to similar MARL algorithms that also involve competitive learning objectives.
>
> > [1-3] are not referenced in this paper.
>
> Thanks, we added these.
>
> >  In fact, instability can still be witnessed in Fig. 3 and I am curious whether changing an RL algorithm would fix such issue.
>
>
> Competitive learning objectives (as described in Eq. $F_{MADDPG}$) naturally give rise to rotational dynamics, which necessitate optimization methods capable of addressing these challenges. While we do not claim that making learning objectives more ‘cooperative’ wouldn’t lead to more stable training, it could potentially reduce efficiency, as competition often facilitates a faster search process. Additionally, not all applications allow for cooperative adjustments. We acknowledge that changes to the RL algorithm itself could further improve performance. However, when competitive objectives are unavoidable, it remains essential to use methods designed to handle the resulting rotational dynamics effectively.
>
>
>
> ## W2: It is unclear whether the property of being closer to equilibrium in simple tasks will bring better performance for more complicated mainstream MARL benchmarks such as SMAC.
>
>
> We demonstrate improvements across multiple tasks where accurate comparisons between methods are possible. Based on our findings, we believe that if the agents’ learning objectives are competitive, the benefits of our approach will likely extend to SMAC as well. Otherwise, we anticipate that the methods will perform no worse than the baseline.

---

> ### Author Response · Authors · 2024-11-25
>
> > Question 1: while it is well-known that MARL with gradient descent is hard to converge, is there any theoretical proof that other VI methods are better than gradient descent when it comes to this special case of VI problems?
>
>
> Yes, under standard assumptions on the agents’ reward functions, such as convexity, the VI becomes monotone, and the methods we consider have convergence guarantees in this case. Thank you for bringing up this question; we have added the paragraph *On the convergence* in Section 4.2 to address your feedback.
>
>
> > Question 2: Can the authors explain why EG does not seem to improve performance as significant as LA, and EG-LA-MADDPG does not work well in the MPE-Predator-prey game and the rock-paper-scissor game in Fig. 5?
>
> Thank you for raising this question. We included a comparison with the widely used Extragradient (EG) method for solving VIs, which is known for its convergence for monotone VIs (unlike GD). However, EG only introduces a minor local adjustment compared to GD. As such, the results align with expectations—while EG occasionally outperforms GD (the baseline), its performance is often similar. In contrast, nested LA applies a significantly stronger contraction, with the degree of contraction increasing as the number of nested levels increases. This leads to substantial performance gains, particularly in terms of stability, as it prevents the last iterate from diverging. However, there is a tradeoff: if the number of nested levels is too high, the steps can become overly conservative or slow. Based on our experiments, three levels of nested LA yielded the best results (see Fig. 1-a).
> It is also worth noting that similar trends have been observed in GANs: while EG performs slightly better than GD, more contractive methods consistently achieve better results. This suggests that the MARL setting involves a substantial rotational component, making it a promising area for further exploration.
> Thank you for pointing out that this is unclear; we have added the above explanation to Section 5.2 in the paragraph *Comparison among [...]* for further clarity.
>
> > Question 3: Is it possible to reach Nash equilibrium by tuning the hyperparameters more? It would be great if the authors could provide more attempts on achieving Nash equilibrium by tuning hyperparameters (which is also an ablation to hyperparameter sensitivity).
>
>
> Thank you for this suggestion. With additional hyperparameter tuning, we observed significantly better performance; please refer to the updated Figure 1(a) for details. These results highlight the critical role of optimization in MARL. While this paper does not aim to address all aspects of the MARL training difficulties, we hope that these findings will encourage further exploration into automated hyperparameter selection specifically for MARL.
> To clarify, we do not claim to fully eliminate hyperparameter sensitivity. However, we show that by employing appropriate optimization methods, the high variability in performance—caused by different random seeds and hyperparameters—is significantly reduced. This is particularly important as the baseline exhibits substantial variance even with only changes to random seeds.

---

> ### Author Response · Authors · 2024-11-25
>
> > Question 4: About "on the rewards as a metric in MARL": the authors claim that we need stronger evaluation metrics in MARL. While I agree that reward cannot fully reflect agent's behavior, I am not sure if other metrics could be called as "stronger" as the goal of multi-agent RL, like any RL, is to maximize reward.
>
>
> In MARL, an agent's rewards depend on the policies learned by other agents—represented as $f_i(x_1, x_2, \dots, x_N)$. This interdependence can result in no discernible difference in rewards between agents that successfully learn to solve a problem and those that do not. For precise examples, please refer to the rock-paper-scissors game in Fig. 3 and the last paragraph of Section 5.2. Consequently, saturated rewards do not indicate that an algorithm has converged to a stable state (because the total sum can be constant while the learned policies rotate in parameter space) or that the overall performance is satisfactory. Furthermore, it is possible for some agents to fail to begin learning altogether.
> This issue is not unique to MARL but also applies to other multi-player machine learning setups. Unlike classification, where lower $f(x)$ loss implies better performance on training data, in games, this is not the case. For example, in GANs, a low generator loss (that depends on the discriminator) does not indicate good performance, which is why external metrics like FID or IS scores are commonly used.
> Thus, while we agree that *the learning objective in multi-agent RL, like any RL, is to maximize reward*, this cannot be used as a performance comparison between methods. Our findings and output inspections suggest the importance of using environments where external metrics can reliably quantify performance. For example, using games where it is possible to assess how effectively the agents solve a given task. A more detailed evaluation should also include measuring whether all agents contribute meaningfully to the task's completion.
>
> ---
> Thank you again for your feedback and time. If your concerns have been addressed, we kindly ask you to consider revising your score. If there are any remaining questions or points requiring clarification, please do not hesitate to let us know—we would be happy to provide additional details.

---

> ### Comment · Reviewer_UjtH · 2024-11-28
>
> Thank you for your detailed response. Here are my follow-up questions:
>
> 1. When I write "I am curious whether changing an RL algorithm would fix such issue", I mean changing MADDPG to newer algorithms (e.g. MATD3), not changing reward function to make the agents cooperative.
>
> 2. I appreciate the authors' effect in introducing results of MATD3, but I feel this paper should more based on MATD3 instead of MADDPG as the latter (like other reviewers pointed out) is outdated. Currently, the story of the paper is still focused on MADDPG (e.g. "We introduce three algorithms that use nLA-VI, EG,and a combination of both, named LA-MADDPG, EG-MADDPG,and LA-EG-MADDPG,respectively" in the abstract).
>
> 3. I feel the "on the convergence" paragraph should be elaborated. More specifically:
>
> a) Why does monotonicity ensures convergence? The paper claims that it is "known" in line 462-463 without reference. Can the authors add related theorems to make it more self-contained?
>
> b) What is the "standard assumption" on the agent's reward functions, and why under this assumption VI becomes monotonic?
>
> 4. The current work does not seem to study scalability where the number of agents are much more than 3; for example, there could be over two dozens of agents in SMAC. Can the proposed method scale when there are more agents in the environment?

---

> > ### Author Response · Authors · 2024-12-04
> >
> > Thanks a lot for engaging in a discussion.
> >
> > **1.** Yes, understood. We point out that our work addresses rotational learning dynamics – one can think of it as dealing with issues on a lower-level (where the algorithmic design is on a higher level). Irrespective of the MARL algorithm, the resulting learning dynamics in multi-agent (and actor-critic) can be purely rotational (highly competitive), purely potential (cooperative) or in between.  Our approach will have reduced benefits solely if the algorithmic design makes the objectives more cooperative.
> >
> > **2.** We agree that, generally, adding more algorithms is beneficial. However, we believe that MADDPG remains highly relevant, as evidenced by our results and current use [e.g., 1, 2, 3].  Importantly, our findings convey a strong message as the improvements we observe across datasets are consistent.
> >
> > **3-a.** The referenced line numbers are from the experiments section, where we discuss our results. Regarding the convergence guarantees, we provide references in the paragraph starting at line 328.
> > To provide intuition, monotonicity for VIs can be thought of as analogous to convexity in optimization. Monotonicity ensures convergence for methods like EG and LA, while (last iterate) gradient descent provably diverges. We can add these guarantees in the appendix where monotonicity is introduced using informal summary theorems.
> >
> > **3-b.** The agents objectives need to be convex, from which assumption the monotonicity of the operator (Def. 1) is implied. To see this connection, one can start from the optimality conditions for convex optimization problems with differentiable objectives for each agent, from which the monotonicity condition follows.
> >
> > **4.** Thanks for the question. Yes, for this response we ran experiments on MPE: Predator-prey environment with 4 adversaries and 2 agents, and our algorithms significantly outperformed the baseline. The results align with our observations in the experiments section for the 2-adversary, 1-agent setting.
> >
> > ---
> > [1]*Multi-Agent Deep Reinforcement Learning-Based Multi-Objective Cooperative Control Strategy for Hybrid Electric Vehicles*, J. Gan, S. Li, X. Lin and X. Tang, 2024.
> >
> > [2]*Time-aware MADDPG with LSTM for multi-agent obstacle avoidance: a comparative study*, Zhao, E., Zhou, N., Liu, C. et al., 2024.
> >
> > [3] *Attention Mechanism-Empowered MADDPG for Distributed Resource Allocation in Cell-Free Mobile Edge Computing*, F. D. Tilahun and C. G. Kang, 2024.

---

### Official Review · Reviewer_pyRk · 2024-10-28

**Soundness:** 2
**Presentation:** 3
**Contribution:** 2
**Rating:** 3
**Confidence:** 3

**Summary:**

The paper investigates the potential of leveraging Variational Inequality (VI)-based techniques to enhance Multi-Agent Reinforcement Learning (MARL) algorithms. Specifically, it incorporates Nested-Lookahead VI and Extra-Gradient into the MADDPG framework and empirically evaluates the modified methods—LA-MADDPG, EG-MADDPG, and LA-EG-MADDPG—on zero-sum games and MPE tasks. The experimental results demonstrate performance improvements over standard optimizers.

**Strengths:**

- The incorporation of VI methods into MARL presents a compelling avenue for research.
- The paper includes detailed pseudocode, enhancing the clarity and understanding of the proposed methods.
- Experimental results indicate improvements compared to traditional methods.

**Weaknesses:**

- The specific problem with MADDPG that the paper addresses is unclear. While the paper discusses VI and its proposed solutions, the alignment with the MARL context and the specific algorithm remains weak. It appears the focus is primarily on a general optimization issue rather than a specialized challenge within MARL.
- The proposed methods seem straightforward, primarily involving a switch from the Adam optimizer to existing VI methods. The motivation for this switch is not adequately explained, and there is a lack of theoretical justification for how LA or EG methods can specifically benefit MADDPG or MARL algorithms.
- The experiments appear somewhat simplistic; a more comprehensive evaluation is necessary. The authors should consider testing in more complex domains and comparing against other MARL methods, as the proposed approach appears generalizable. Additionally, given the limited state-action space in tasks like rock-paper-scissors, theoretical justifications for these tasks would strengthen the argument regarding the shortcomings of existing methods and how the proposed approach addresses them.

**Questions:**

In Figures 2 and 6, the evaluation episodes provide limited information. I would appreciate seeing the learning curves for the two MPE experiments.

---

> ### Author Response · Authors · 2024-11-25
>
> Thank you for taking the time to review our paper and for providing valuable feedback.
>
> > W1: The specific problem with MADDPG that the paper addresses is unclear. It appears the focus is primarily on a general optimization issue rather than a specialized challenge within MARL.
>
> Our work focuses on a general MARL optimization issue rather than a specialized problem specific to a MARL algorithm, and it does not involve introducing changes to the reward functions. Instead, we demonstrate that by improving the optimization—while keeping all other components fixed—we achieve significant gains in both algorithmic stability across random seeds and performance, measured as the distance to equilibrium. For example, please refer to Figure 1 for evidence of these improvements. We view this as a key strength of our work, as it targets a fundamental challenge in MARL. This positions our approach uniquely, bridging concepts from two areas of machine learning to address a critical issue affecting both MARL research progress and deployment.
>
> In response to this feedback, we have updated the introduction to better highlight our motivation and added references to emphasize the significance of the problem and its impact on MARL research and applications.
>
>
> > W2: The proposed methods seem straightforward, primarily involving a switch from the Adam optimizer to existing VI methods. The motivation for this switch is not adequately explained, and there is a lack of theoretical justification for how LA or EG methods can specifically benefit MADDPG or MARL algorithms.
>
> Our work represents an initial step toward establishing a foundation for further development of VI methods in MARL. Our results highlight significant performance gains, demonstrating their timely relevance for MARL challenges.
>
> The VI perspective we present frames MARL as a VI instance, revealing that it generally involves rotational learning dynamics. When addressed with standard optimization methods, these dynamics can lead to training instabilities. For theoretical justification of the proposed methods under standard assumptions, please refer to the paragraph *On the convergence in Section 4.2*. Please let us know if further clarifications could be helpful.
>
>
> > W3: simplistic experiments
>
> We demonstrate improvements across multiple tasks where accurate comparisons between methods are possible using distance to the equilibrium. Our findings, consistent across the experiments, imply that if the agents’ learning objectives are competitive, the benefits of our approach will extend to other benchmarks as well.
>
>
> > Question: In Figures 2 and 6, the evaluation episodes provide limited information. I would appreciate seeing the learning curves for the two MPE experiments.
>
> Thank you for the question. We added those results, please refer to figures $8$ and $9$ and section A.3.4 for the learning curves of MPE experiments.
>
>
> ---
> Thanks again for your feedback. Please consider increasing your score if your concerns are resolved. Otherwise, please let us know and we will be happy to provide further clarifications.

---

### Official Review · Reviewer_nNmJ · 2024-11-03

**Soundness:** 1
**Presentation:** 2
**Contribution:** 2
**Rating:** 3
**Confidence:** 3

**Summary:**

This paper proposes incorporating variational inequality methods into multi-agent reinforcement learning to address its sensitivity to hyperparameters caused by gradient-based updates. The authors use two types of VIs on top of MADDPG and evaluate them alongside MADDPG.

**Strengths:**

- Sensitivity to hyperparameters is a significant problem in reinforcement learning, impacting its practical applications.
- Introducing variational inequalities to improve stability in finding equilibrium is an interesting approach.

**Weaknesses:**

The weakest part of this paper is the motivation/results section.

- The authors introduce VIs to tackle the high sensitivity to hyperparameters in RL. However, there are no experiments provided that demonstrate the proposed method’s effectiveness in addressing this issue. I recommend that the authors illustrate the extent of MADDPG’s sensitivity to hyperparameters, identifying which hyperparameters are particularly sensitive (an analysis of high sensitivity), and then show how effectively the proposed method stabilizes this sensitivity compared to MADDPG. Including additional analysis and ablation studies would further verify this claim. Based on the current results, it is difficult to conclude that the proposed method addresses the problem the authors aim to solve.


- The proposed method primarily relies on MADDPG, but it would be beneficial if the authors provided results of the proposed method applied to other algorithms (including SOTA, as MADDPG is somewhat outdated). This would help demonstrate the generalizability of the method.

- Figure 2 does not effectively illustrate the benefits of the proposed method.

**Questions:**

See weaknesses

---

> ### Author Response · Authors · 2024-11-25
>
> Thanks a lot for your time reviewing our paper and for your feedback.
>
> > W1-motivation: high sensitivity to hyperparameters in RL
>
> To clarify, our work does not aim to address hyperparameter tuning or the elimination of hyperparameters. Instead, our motivation lies in tackling the broader challenges of MARL training, commonly referred to as the MARL reproducibility crisis [1]. These challenges include:
> - The well-documented difficulty of MARL training, where convergence may fail to initiate.
> - Significant performance variability caused by small changes in hyperparameters or initial random seeds.
> Our results address challenge (1) and partially mitigate (2), as the variance across random seeds is notably reduced. Furthermore, the comparison of the proposed VI methods across benchmarks shows consistent and expected improvements; see the updated paragraph *Comparison among [...]* in Section 5.2 for more details.
>
> Thank you for highlighting this point. We have updated the introduction to better address these aspects; please see Section 1, particularly the second paragraph.
>
>
> > W2: [....] it would be beneficial if the authors provided results of the proposed method applied to other algorithms
>
> We compared MADDPG with MATD3 (Ackermann et al., 2019); please see Figure $10$  in Appendix A.4 for details. Our observations show that the performance of both algorithms fluctuates between high and low rewards, indicating similar behavior. This suggests that our proposed methods, which address rotational dynamics, would be beneficial for both algorithms. Our experiments demonstrate relative improvements over the baseline. Therefore, the insights gained from our work are applicable to similar MARL algorithms that also involve competitive learning objectives.
>
>
> > W2: Figure 2 does not effectively illustrate the benefits of the proposed method.
>
> For the *predator-prey* benchmark, rewards are not indicative of performance as they depend on the policies of other agents (see *On the rewards as a metric in MARL* in Section 5.2). Instead, we use the win rate of one team as a metric in Figure 2 and provide reward values and algorithm snapshots in the appendix.
>
> *Metric in Fig. 2.* If all agents develop strong policies, the adversary team’s win rate against the agent should approach $0.5$, as the agent has a speed advantage, but coordinated adversaries can improve their chances of capturing it. A win rate closer to $0.5$ indicates better adversarial coordination, observed with LA-MADDPG. In contrast, baseline MADDPG often results in only one adversary learning to chase, while the other moves away or remains stationary, leading to a lower win rate due to poor coordination.
>
> *Other metrics in Appendix.*  Visual snapshots in Figure 7 show that baseline MADDPG often diverges, moving away from targets, which highlights suboptimal behavior. Figure 8 provides the rewards during training for completeness.
>
>
> ---
>
> [1] *BenchMARL: Benchmarking Multi-Agent Reinforcement Learning*, Bettini, Prorok, Moens, 2024.
>
> ---
> Thank you again for your valuable feedback. If we addressed your concerns, please consider revising your score. Otherwise, if there are any remaining questions or clarifications needed, please let us know.

---

### Official Review · Reviewer_3iDU · 2024-11-04

**Soundness:** 1
**Presentation:** 3
**Contribution:** 1
**Rating:** 3
**Confidence:** 4

**Summary:**

The paper investigates combining multi-agent policy gradients with game-theoretical gradients based on variational inequalities (VIs). It examines whether incorporating VI-based optimization methods can enhance policy optimization within MARL frameworks.

In practical terms, the authors propose an extension to the Multi-Agent Deep Deterministic Policy Gradient (MADDPG) method. Rather than employing the standard MADDPG gradient updates, the paper introduces modifications using extragradient and nested-lookahead techniques to adjust the MADDPG gradients.

**Strengths:**

1. The paper is very well organized and easy to follow.

2. VI methods and MAPG methods, though highly related, are somehow separated in the current multi-agent literature. The reviewer was aware of some papers discussing that MAPG cannot converge to NEs. But generally, in the existing literature, the default context for MAPG is cooperative tasks, while VI methods are for zero-sum games--leading to a gap.

**Weaknesses:**

1. The __motivation__ of the proposed method needs significant improvement.

It is strange to motive the paper by the hyper-parameter sensitivity observed in MARL. The rotational dynamics of gradient-based methods in multi-agent games is inherent and cannot be eliminated by hyper-parameter fine-tuning. This is well studied in the literature, and is known to be attributable to the asymmetric component of the Jacobian of the limiting ordinary differential equations (ODE).

Contrary to the authors' implication, combining MAPG (multi-agent policy gradient) methods with VI cannot, by itself, address the motivating problems mentioned by the authors. To be specific, they are still very sensitive to hyper-parameters. Many VI methods like sympletic gradient adjustment and consensus optimization even introduce more hyper-parameters, for the purpose of addressing the asymmetric component in the Jacobian. Moreover, the reviewer also didn't see how the proposed method is able to improve the search space of MAPG, which is another motivating problem of this paper.

2. The application of extragradient methods, as referenced in seminal work [1], is typically confined to coherent saddle point problems where all saddle points are assumed to be local Nash equilibria—an assumption not generally applicable in MARL environments.

This discrepancy raises concerns about the applicability of the proposed methods to the broader class of problems presented in MARL scenarios. Without addressing these fundamental theoretical mismatches, there is no assurance that the proposed method can effectively converge to differential Nash Equilibria, which the paper identifies as its solution concept.

3. The __results__ of the method are not informative.

The presentation of the results lacks clarity and does not convincingly demonstrate the efficacy of integrating MADDPG with VI techniques.

The experiments, as depicted, do not show meaningful improvement in learning outcomes. What is point of using MADDPG + VI in games? Most advanced VI methods can already find the NEs of these games.

In Fig. 2,  the proposed method does not show any improvements during learning. Why?

Moreover, the high variance observed in the results casts further doubt on the method's reliability and robustness. Such discrepancies point to a need for more rigorous experimental design and a deeper analysis to genuinely assess the method's impact on performance in MARL settings.

[1] Mazumdar, E.V., Jordan, M.I. and Sastry, S.S., 2019. On finding local nash equilibria (and only local nash equilibria) in zero-sum games. arXiv preprint arXiv:1901.00838.

**Questions:**

See the previous section.

---

> ### Author Response · Authors · 2024-11-25
>
> ## W2: controlling for NE convergence
>
>
>
> Thank you for your perspective. While we understand your point, theoretical guarantees, as in GANs and deep learning, often do not fully apply to general neural networks. Our approach builds on recent VI optimization advancements, highlighting the role of rotational dynamics in MARL. Please see also paragraph *On the convergence* in Section 4.2 of the updated version.  While we acknowledge Mazumdar et al., 2019, proposing a second-order method, such approaches are computationally prohibitive. Hence, we focus on first-order VI methods that are more practical yet effectively address rotational dynamics.
>
>
> ## W3: results
>
> > The presentation of the results lacks clarity and does not convincingly demonstrate the efficacy of integrating MADDPG with VI techniques.
>
> Thank you for raising this point. Our results consistently show improvements in equilibrium distance and reduced variance across random seeds. In summary:
>
> - The baseline (in blue) exhibits high variance across initial random seeds and often diverges.
> - EG, with only minor differences from the baseline, shows slight improvements but not always significant.
> - Nested LA, which introduces a stronger contraction, achieves significant performance gains.
>
> Therefore, we respectfully disagree with the above assessment, as our findings provide strong evidence of the impact of the proposed optimization methods on MARL performance. Figure 1-a illustrates that simply improving the optimization process leads to substantial performance gains.
>
> Further details on the consistent outperformance of the methods across benchmarks are provided in Section 5.2, specifically in the paragraph *Comparison among [...]*. We added the paragraph *Summary* in Section 5.2 in response to your feedback on the confusion. Please let us know if additional clarification would be helpful.
>
> > What is point of using MADDPG + VI in games?
>
>
> As mentioned above, we leverage VI-solving methods to address the rotational dynamics inherent in MARL training, which traditional gradient-based approaches struggle to handle effectively. This focus is core to our methodology, and we would be happy to provide further clarification if needed.
>
> > In Fig. 2, the proposed method does not show any improvements during learning. Why?
>
> This behavior is expected due to the metric used. The $y$-axis in Fig. 2 represents the average win rate of adversarial agents. While both teams are improving their policies over time, the plotted win rate reflects performance relative to the opposing team at a given iteration ($x$-axis). As both teams learn and adapt simultaneously, the win rate fluctuates, even as the overall quality of policies improves.
>
>
> > Moreover, the high variance observed in the results casts further doubt on the method's reliability and robustness. Such discrepancies point to a need for more rigorous experimental design and a deeper analysis to genuinely assess the method's impact on performance in MARL settings.
>
> We respectfully believe there may be a misunderstanding here. The high variance observed in the baseline results is precisely the motivation for our work, as this issue is well-documented in MARL literature (discussed in Sec. 1). Our methods aim to mitigate this variance by addressing the underlying rotational dynamics, which we show is a key challenge in MARL. Our approach has demonstrated significant improvements in stability and performance across benchmarks.
>
> ---
> [1] *BenchMARL: Benchmarking Multi-Agent Reinforcement Learning*, Bettini, Prorok, Moens, 2024.
>
> [2] *Last-Iterate Convergence of Saddle Point Optimizers via High-Resolution Differential Equations*, Chavdarova, Jordan, Zampetakis, 2023.
>
> ---
> We would like to thank the reviewer again for their comments and time. Please consider increasing your score if you believe your concerns have been resolved. Otherwise, please let us know and we will be happy to provide further clarifications.

---

### Meta-Review · Area_Chair_bRLL · 2024-12-11

**Metareview:**

This paper incorporates variational inequality methods, in particular, Nested-Lookahead VI and Extra-Gradient, into multi-agent reinforcement learning (MARL) to address its sensitivity to hyperparameters caused by gradient-based updates. Experiments were provided to demonstrate the effectiveness of the variants. The overall idea is new, and the paper is generally easy to follow. However, there were some concerns regarding the motivation (e.g., how much does MADDPG suffer from the sensitivity to hyperparameters), limitation of the general applicability of the methods to MARL algorithms other than MADDPG, the benefits of using VI-based methods specific to MARL (in contrast to general optimization), as well as the insufficiency of the experiments. Some theoretical analyses would also strengthen the soundness of the paper. I suggest the authors incorporate the feedback in preparing the next version of the paper.

**Additional Comments On Reviewer Discussion:**

There were some concerns regarding the general applicability of the methods, the motivation for the use of VI-based methods, the simplicity of the experiments, and the lack of theoretical analyses. The authors' responses addressed part of the concerns, but not fully, and there are still issues left unaddressed (as mentioned above).

---

### Decision · Program_Chairs · 2025-01-22

Reject